# LoTA-QAF: Lossless Ternary Adaptation for Quantization-Aware Fine-Tuning

**Junyu Chen**[1,2,3], **Junzhuo Li**[3], **Zhen Peng**[4], **Wenjie Wang**[1,2],
**Yuxiang Ren**[5], **Long Shi**[1,2,*], **Xuming Hu**[3,*]

[1] Southwestern University of Finance and Economics
[2] Artificial Intelligence and Digital Finance Key Laboratory of Sichuan Province
[3] The Hong Kong University of Science and Technology (Guangzhou)
[4] Sun Yat-sen University      [5] Nanjing University

223081200039@smail.swufe.edu.cn
shilong@swufe.edu.cn    xuminghu@hkust-gz.edu.cn

## Abstract

Quantization and fine-tuning are crucial for deploying large language models (LLMs) on resource-constrained edge devices. However, fine-tuning quantized models presents significant challenges, primarily stemming from: First, the mismatch in data types between the low-precision quantized weights (e.g., 4-bit) and the high-precision adaptation weights (e.g., 16-bit). This mismatch limits the computational efficiency advantage offered by quantized weights during inference. Second, potential accuracy degradation when merging these high-precision adaptation weights into the low-precision quantized weights, as the adaptation weights often necessitate approximation or truncation. Third, as far as we know, no existing methods support the lossless merging of adaptation while adjusting all quantized weights. To address these challenges, we introduce lossless ternary adaptation for quantization-aware fine-tuning (LoTA-QAF). This is a novel fine-tuning method specifically designed for quantized LLMs, enabling the lossless merging of ternary adaptation weights into quantized weights and the adjustment of all quantized weights. LoTA-QAF operates through a combination of: i) A custom-designed ternary adaptation (TA) that aligns ternary weights with the quantization grid and uses these ternary weights to adjust quantized weights. ii) A TA-based mechanism that enables the lossless merging of adaptation weights. iii) Ternary signed gradient descent (t-SignSGD) for updating the TA weights. We apply LoTA-QAF to Llama-3.1/3.3 and Qwen-2.5 model families and validate its effectiveness on several downstream tasks. On the MMLU benchmark, our method effectively recovers performance for quantized models, surpassing 16-bit LoRA by up to 5.14%. For task-specific fine-tuning, 16-bit LoRA achieves superior results, but LoTA-QAF still outperforms other methods. Code is available in github.com/KingdalfGoodman/LoTA-QAF.

## 1   Introduction

Large language models (LLMs) have showcased exceptional proficiency in natural language processing, driving advancements in applications such as complex reasoning (Hadi et al., 2023), code generation (Jiang et al., 2024), and conversational AI systems (McTear and Ashurkina, 2024). However, their immense computational costs present significant challenges, particularly when deploying

---

* Corresponding authors.

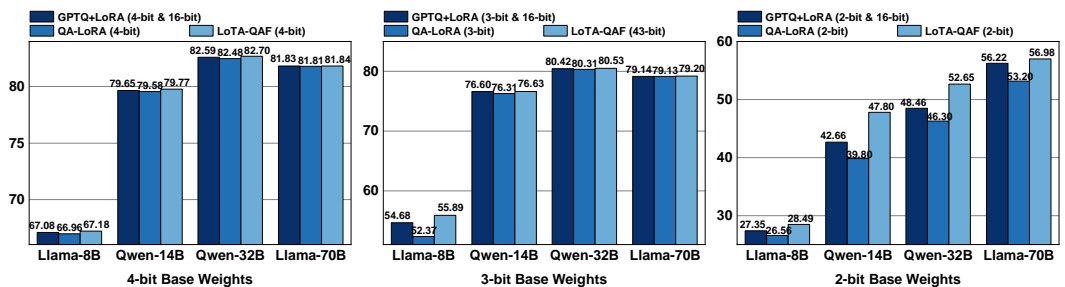

Figure 1: Comparison of 5-shot MMLU accuracy (%), fine-tuned on the Alpaca dataset. GPTQ+LoRA (4-bit & 16-bit) uses 4-bit base and 16-bit adapter weights. QA-LoRA and LoTA-QAF operate at 4-bit after merging adapters into quantized weights. Details in Table 1.

these models on resource-constrained edge devices (Wan et al., 2023). To overcome this hurdle, quantization techniques have emerged as a critical approach to model compression, effectively reducing memory demands and computational complexity by quantizing model weights to lower bit precision. Furthermore, deploying on edge devices not only necessitates model compression but also requires the models to handle specialized tasks, which require the integration of domain-specific knowledge. Consequently, there is growing attention on fine-tuning LLMs to align with specific tasks under quantization constraints. One prominent approach designed for this scenario is QLoRA (Dettmers et al., 2023), which integrates low-rank adaptation with quantization. It quantizes the pretrained weights to 4-bit precision, significantly reducing the static memory footprint. Subsequently, 16-bit precision LoRA adapters are trained on top of these quantized weights, achieving efficient and low-cost fine-tuning.

Nevertheless, directly applying adapters to fine-tuning quantized models introduces notable challenges that can hinder efficiency and performance. The first challenge arises during inference, where the interaction between low-precision weights (e.g., 4-bit) and high-precision adapter weights (e.g., 16-bit) can impair computational efficiency. This mismatch introduces computational overhead that can diminish the inference speedups from quantization (Jeon et al., 2024). The second challenge occurs when merging high-precision LoRA adapters into the low-precision quantized weights. During this process, the adapters must inevitably be quantized or truncated to the lower bit width (Bondarenko et al., 2024; Guo et al., 2024). This reintroduction of quantization error at the adapter level, despite the adapters' aim to correct model quantization errors, leads to fine-tuning accuracy degradation. The third challenge is that existing lossless adapter merging schemes (Xu et al., 2023) only allow for adjustment of quantization parameters, rather than direct modification of the quantized weights. This significantly limits the adapter's capacity for effective fine-tuning.

These challenges highlight the development of fine-tuning methods integrated within a quantization framework, namely quantization-aware fine-tuning (QAF). In this paper, we present LoTA-QAF, a lossless ternary adaptation fine-tuning method for quantized pretrained language models. Specifically, LoTA-QAF operates through a combination of three key components: i) A custom-designed ternary adaptation (TA) that includes trainable ternary adapters, which form an auxiliary matrix to adjust the quantized weights within the target quantization grid, instead of merely adjusting quantization parameters. ii) A TA-based lossless merging mechanism that utilizes the auxiliary matrix to generate a ternary matrix and an offset matrix, subsequently updating the quantized weights and zero factors. This mechanism preserves the low-bit computational efficiency and avoids fine-tuning accuracy degradation. iii) Ternary signed gradient descent (t-SignSGD) is employed to update the trainable ternary adapters during fine-tuning.

We apply LoTA-QAF to the Llama-3.1/3.3 and Qwen-2.5 model families, validating its effectiveness in performance recovery and task-specific alignment for quantized models. Figure 1 illustrates the MMLU benchmark results, demonstrating that LoTA-QAF consistently outperforms other methods, especially at 2-bit. It should be noted that LoRA employs 16-bit adapters (inference with 4-bit model weights and 16-bit adapters), which cannot be losslessly merged into quantized weights. In contrast, both LoTA-QAF and QA-LoRA can achieve lossless merging (inference with only 4-bit model weights). However, QA-LoRA only indirectly compensates for quantization errors by adjusting zero

factors via its adapter, lacking the capability to directly modify the quantized weights. Building on this, our main contributions include:

- We present LoTA-QAF, which adjusts quantized weights within the quantization grid and facilitating the lossless merging of ternary adapters into the quantized weights. As a result, LoTA-QAF preserves low-bit computational efficiency and avoids accuracy degradation.

- We introduce ternary signed gradient descent (t-SignSGD), a novel optimizer for ternary adapters. It leverages sign-based updates with dynamic percentile-based thresholding to selectively adjust ternary adapter weights, thereby effectively optimizing these highly constrained parameters.

- LoTA-QAF demonstrates effectiveness in two key quantization-aware fine-tuning scenarios, achieving strong performance while maintaining the inference efficiency of quantized models. For performance recovery, LoTA-QAF shows up to 5.14% improvement over LoRA on Qwen 2.5 14B 2-bit. In task-specific tuning, while trailing LoRA (16-bit adapters), LoTA-QAF still outperforms other methods. Regarding inference efficiency, LoTA-QAF is 1.7x-2.0x faster than LoRA after adapters are merged into quantized weights.

## 2 Related Work

**Quantization of LLMs.** Quantization techniques fundamentally reduce memory requirements and improve computational efficiency by lowering the bit precision of model weights. However, the precision loss inherent in quantization inevitably leads to a decline in the performance of quantized models (Kumar et al., 2024). Consequently, recovering the performance of quantized models is a crucial research topic in the field. Post-training quantization (PTQ) offers a fast compression solution without requiring retraining. One approach, GPTQ (Frantar et al., 2022), employs approximate second-order information during the quantization process, compensating for the quantization errors progressively. In addition, other approaches aim to address outliers, such as AWQ (Lin et al., 2024), which identifies and protects salient weights by scaling channels based on activation statistics. QuaRot (Ashkboos et al., 2024) uses randomized Hadamard transformations to remove outliers from the hidden state. Nevertheless, PTQ cannot adapt models to downstream tasks because it is applied after training.

Quantization-aware training (QAT), which integrates quantization simulation during training (Chen et al., 2024; Liu et al., 2023), not only helps performance recovery but also aligns quantized models with specific tasks. LLM-QAT (Liu et al., 2023) introduces QAT for LLMs using a data-free distillation approach to preserve the original model's output distribution. Efficient-QAT (Chen et al., 2024) proposes a two-stage strategy involving block-wise training of all parameters and end-to-end training of quantization parameters. HALO (Ashkboos et al., 2025) introduces Hadamard rotations during both forward and backward passes to mitigate outliers in low-precision training. However, the computational cost of QAT remains a significant challenge.

**Quantization-Aware Fine-Tuning of LLMs.** To maintain the low computational cost of quantization, while achieving performance recovery and alignment with specific tasks through fine-tuning, Bondarenko et al. (2024); Dettmers et al. (2023); Xu et al. (2023) explore quantization-aware fine-tuning (QAF) of large language models (LLMs). Specifically, the pivotal work, QLoRA (Dettmers et al., 2023), employs fine-tuning of LoRA adapters over a frozen quantized model. LoftQ (Li et al., 2023b) finds an optimized low-rank initialization for adapters. RoLoRA (Huang et al., 2024) incorporates rotation mechanisms during LoRA fine-tuning over quantized weights to mitigate outlier. LR-QAT (Bondarenko et al., 2024) trains adapters within the QAT framework, making the adapters quantization-aware. QA-LoRA (Xu et al., 2023) focuses on enabling a lossless merge of the trained adapter into the quantized weights. It achieves this by structuring the adapter to align with group-wise quantization parameters, allowing the adapter's influence to be absorbed into the zero factors.

Based on our literature review, the closest work to ours is QA-LoRA, which achieves lossless merging of adapters. However, QA-LoRA only adjusts the zero factors in group-wise quantization using the adapter, which restricts its fine-tuning capability. In contrast, our LoTA-QAF allows fine-tuning of all quantized weights within the quantization grid while maintaining the lossless merge property.

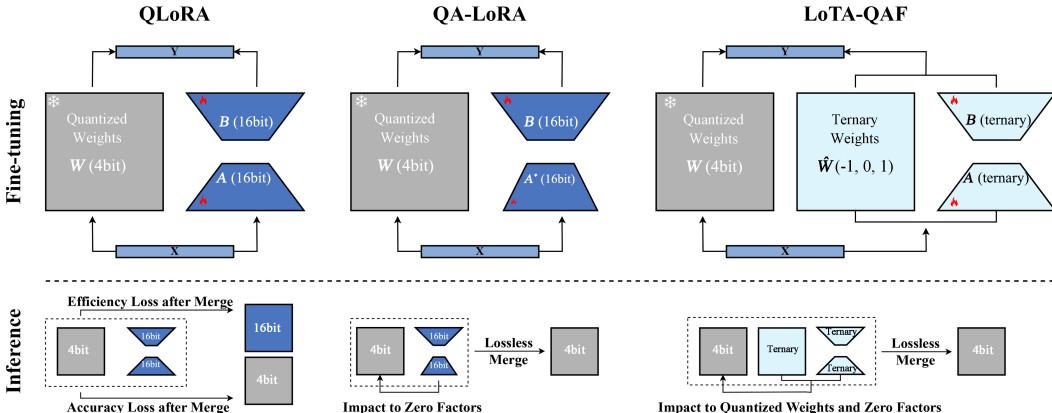

Figure 2: Comparison of prior adaptation methods with LoTA-QAF. A merge can result in efficiency loss, where model weights are de-quantized to preserve the adapter's precision but sacrifice inference efficiency, or accuracy loss, where the adapter's precision is sacrificed to maintain low-bit inference. A lossless merge avoids this trade-off because adapter weights are utilized identically during both the training and inference (after merge), ensuring the adapter's learned effect is fully preserved.

## 3 Method

### 3.1 Preliminaries

**Low-Rank Adaptation.**  The pre-trained weight matrix $\mathbf{W} \in \mathbb{R}^{D_{\text{in}} \times D_{\text{out}}}$ is frozen during the fine-tuning process, where $D_{\text{in}}$ and $D_{\text{out}}$ represent the input and output dimensions, respectively. To enable efficient fine-tuning, trainable adapters $\mathbf{A} \in \mathbb{R}^{D_{\text{in}} \times r}$ and $\mathbf{B} \in \mathbb{R}^{r \times D_{\text{out}}}$ are introduced, where the rank $r \ll \min(D_{\text{in}}, D_{\text{out}})$. These adapters are multiplied to form a low-rank matrix $\mathbf{AB}$, which fine-tuning model with significantly fewer trainable parameters. The modified forward pass is computed as

$$\mathbf{y} = (\mathbf{W} + \frac{\alpha}{r}\mathbf{AB})^T\mathbf{x}, \tag{1}$$

where $\mathbf{A}$ is initialized with random Gaussian distribution, while $\mathbf{B}$ is initialized to zero. Here, $\alpha$ is a constant scaling factor that adjusts the magnitude of the low-rank update.

**Asymmetric Affine Quantization.**  We perform $N$-bit affine quantization on the weight matrix $\mathbf{W}$, the quantization process maps $\mathbf{W}$ to an integer matrix $\mathbf{W}_{\text{int}}$. The dequantized approximation $\mathbf{W}_{\text{q}}$ as follows:

$$\mathbf{W}_{\text{q}} = s\mathbf{W}_{\text{int}} + z = s\left\lfloor \frac{\mathbf{W} - z}{s} \right\rceil + z, \tag{2}$$

where the scaling factor $s = (\max(\mathbf{W}) - \min(\mathbf{W}))/(2^N - 1)$ and the zero factor $z = \min(\mathbf{W})$. The symbol $\lfloor \cdot \rceil$ denotes rounding to the nearest integer. All elements in the integer matrix $\mathbf{W}_{\text{int}}$ belong to the set $\{0, 1, \ldots, 2^N - 1\}$.

### 3.2 Lossless Ternary Adaptation

When fine-tuning quantized models using low-rank adaptation to compensate for quantization loss or adapt to a specific task, the key aspect is adjusting quantized weights $\mathbf{W}_{\text{int}}$. Moreover, the merge process should not quantize or truncate the adapter weights, which would reintroduce quantization errors at the adapter level. Consequently, we design the ternary adaptation, which losslessly merges the ternary adaptation into the quantized weights. This adaptation utilizes trainable ternary adapters $\mathbf{A}_{\text{T}} \in \{-1, 0, 1\}^{D_{\text{in}} \times r}$ and $\mathbf{B}_{\text{T}} \in \{-1, 0, 1\}^{r \times D_{\text{out}}}$. Consistent with low-rank adaptation principles, the rank $r \ll \min(D_{\text{in}}, D_{\text{out}})$. For initialization, we apply Kaiming normal initialization (He et al.,

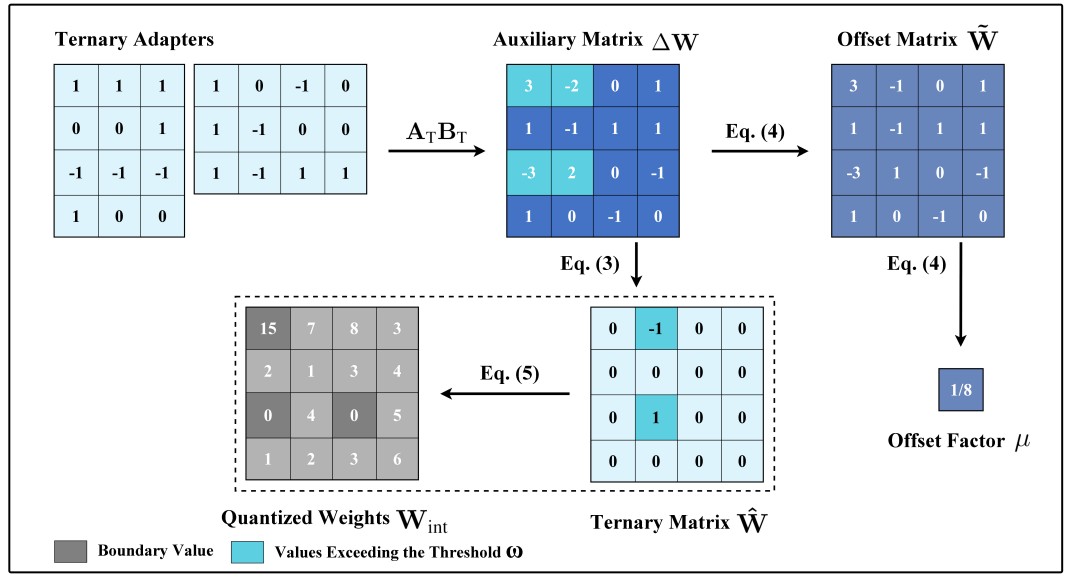

Figure 3: Illustration of LoTA-QAF. The process is demonstrated using a small 4x4 matrix under 4-bit quantization, with adapter rank $r$ set to 3 and threshold $\omega$ to 1. Notably, quantized weights have boundary values (e.g., 0 and 15 for 4-bit, where 15 cannot be incremented and 0 cannot be decremented), requiring overflow prevention.

2015) to $\mathbf{A}_{\mathrm{T}}$ and then ternarize its elements using a threshold, calculated as 0.75 times the mean absolute value of the sampled weights (Li et al., 2016), while $\mathbf{B}_{\mathrm{T}}$ is initialized to zeros.

Notably, the product of the ternary adapters $\mathbf{A}_{\mathrm{T}}$ and $\mathbf{B}_{\mathrm{T}}$ forms the auxiliary matrix $\Delta \mathbf{W} = \mathbf{A}_{\mathrm{T}} \mathbf{B}_{\mathrm{T}}$. Its elements $\Delta \mathbf{W}_{ij}$ are integers within the range $[-r, r]$, where $i \in \{1, \ldots, D_{\mathrm{in}}\}$ and $j \in \{1, \ldots, D_{\mathrm{out}}\}$. The auxiliary matrix $\Delta \mathbf{W}$ is mapped to the ternary matrix $\hat{\mathbf{W}} \in \{-1, 0, 1\}^{D_{\mathrm{in}} \times D_{\mathrm{out}}}$ using a threshold $\omega \in (0, r)$, as follows:

$$\hat{\mathbf{W}}_{ij} = \operatorname{sign}(\Delta \mathbf{W}_{ij}) \cdot \mathbb{I}_{|\Delta \mathbf{w}_{ij}| > \omega}, \tag{3}$$

where $\operatorname{sign}(\cdot)$ denotes the sign function, and $\mathbb{I}_{|\Delta \mathbf{w}_{ij}| > \omega}$ represents the indicator function, which evaluates to 1 if $|\Delta \mathbf{W}_{ij}| > \omega$ and 0 otherwise. The resulting ternary matrix $\hat{\mathbf{W}}$ has the same dimensions as the quantized weight matrix $\mathbf{W}_{\mathrm{int}}$. Consequently, $\hat{\mathbf{W}}$ is used to adjust the quantized weights $\mathbf{W}_{\mathrm{int}}$, compensating for quantization errors or adapting the model to specific tasks. Additionally, $\hat{\mathbf{W}}$ can be losslessly merged into $\mathbf{W}_{\mathrm{int}}$ after fine-tuning, preserving the inference efficiency of the quantized weights. We provide further analysis of the threshold $\omega$ in Section 4.3.

Furthermore, an offset factor $\mu$ is introduced, which represents an average offset relative to $\Delta \mathbf{W}$, and ultimately absorbed into the zero factor. The calculation of $\mu$ relies on the offset matrix $\tilde{\mathbf{W}}$. This offset matrix $\tilde{\mathbf{W}}$ and the offset factor $\mu$ are then computed as:

$$\begin{aligned}
\tilde{\mathbf{W}}_{ij} &= \Delta \mathbf{W}_{ij} - \omega \hat{\mathbf{W}}_{ij} \\
\mu &= \frac{\sum_{i=1}^{D_{\mathrm{in}}} \sum_{j=1}^{D_{\mathrm{out}}} \tilde{\mathbf{W}}_{ij}}{D_{\mathrm{in}} D_{\mathrm{out}}},
\end{aligned} \tag{4}$$

where offset factor $\mu$ can be performed at different granularity, such as per-tensor, per-group, or per-channel, depending on the specific quantization method employed.

Following the fine-tuning stage, the lossless ternary adaptation merge mechanism is defined as:

$$\begin{aligned}
\mathbf{W}'_{\mathrm{int}} &= \mathbf{W}_{\mathrm{int}} + \hat{\mathbf{W}} \\
z' &= z + s\mu
\end{aligned} \tag{5}$$

where the final weights $\mathbf{W}'_{\mathrm{int}}$ belong to the set $\{0, 1, \ldots, 2^N - 1\}$. Equivalently, the forward pass of the unmerged module is defined as $\mathbf{y} = (s \cdot \mathbf{W}'_{\mathrm{int}} + z')^T \mathbf{x}$. Therefore, the lossless ternary adaptation

preserves low-bit computational efficiency and avoids the reintroduction of quantization loss at the adapter level.

## 3.3 Ternary Signed Gradient Descent

Inspired by the application of signed gradient descent (SignSGD) for quantization-related parameter updates in AutoRound (Cheng et al., 2023), we propose t-SignSGD for optimizing ternary adapters. Indeed, Balles et al. (2020); Li et al. (2024, 2023a); Safaryan and Richtárik (2021) demonstrate that SignSGD is beneficially applied to updates within discrete and constrained domains. This characteristic is highly relevant to our ternary adapters, which are constrained to the discrete values $\{-1, 0, 1\}$. Specifically, our proposed t-SignSGD performs updates on the ternary adapters (e.g., $\mathbf{A}_\text{T}$). Let $g_t = \nabla_{\mathbf{A}_\text{T}} \mathcal{L}|_{\mathbf{A}_{\text{T},t}}$ denote the gradient at iteration $t$. To focus updates on prominent gradients, we employ a fixed minimum gradient threshold $\tau$ (e.g., $1 \times 10^{-9}$) and a dynamic percentile-based threshold $\sigma_t$. The threshold $\sigma_t$ is determined at each iteration $t$ based on a specific percentile of the gradient magnitudes, selecting a certain proportion of the largest gradients for update. This proportion of selected gradients is initially the top 5% and then linearly decays to 0.01% during training. The t-SignSGD update is performed as follows:

$$\mathbf{A}_{\text{T},t+1} = \text{clip}\left(\mathbf{A}_{\text{T},t} - \text{sign}(g_t) \cdot \mathbb{I}_{|g_t| > \max(\tau, \sigma_t)}, -1, 1\right), \tag{6}$$

where the indicator function $\mathbb{I}_{(\cdot)}$ equals 1 if the condition ($|g_t| > \max(\tau, \sigma_t)$) is satisfied, and 0 otherwise. The clip$(\cdot, -1, 1)$ function ensures that the updated adapter weights $\mathbf{A}_{\text{T},t+1}$ are strictly constrained to ternary set $\{-1, 0, 1\}$. Notably, Eq. (6) does not use a learning rate to scale the sign of the gradient. Instead, if a ternary adapter weight $\mathbf{A}_\text{T}$ is selected for an update (i.e., its gradient magnitude $|g_t|$ exceeds the threshold $\max(\tau, \sigma_t)$), its value is adjusted by $-\text{sign}(g_t)$. Therefore, the threshold $\sigma_t$ play a crucial role in determining the *selectivity* of updates. By targeting only the top percentile of gradient magnitudes (e.g., the top 5%), $\sigma_t$ acts as an adaptive mechanism that focuses updates on the most salient ternary adapter weights at each iteration $t$.

The convergence of t-SignSGD is primarily governed by its update rule, which is designed for the discrete and bounded nature of the ternary adapters. The key component is the indicator function $\mathbb{I}_{|g_t| > \max(\tau, \sigma_t)}$ in Eq. (6), which acts as an adaptive and dynamic selection mechanism. It ensures stable convergence in two ways: i) Stability via noise filtering (preventing oscillation). In discrete optimization, small, noisy gradients can cause parameters to oscillate unstably between states (e.g., flipping between 0 and 1). Our thresholding mechanism filters out these low-magnitude gradients, ensuring that updates are only performed when the gradient signal is strong and reliable. This addresses the core challenge in sign-based methods related to low signal-to-noise ratios, effectively preventing unstable "jitter" and promoting a smoother convergence path. ii) Convergence via annealing-like dynamics. The decaying percentile threshold $\sigma_t$ introduces an annealing-like, coarse-to-fine search strategy. Early in training ($\sigma_t$ is high), the top percentile of gradients triggers updates. This focuses the optimization on the most impactful parameters, allowing the model to quickly perform "broad-stroke" adjustments and find a promising region in the vast solution space. Later in training ($\sigma_t$ is low), allowing for more fine-grained adjustments. This enables the model to refine its solution within the promising region, analogous to the exploitation phase in classic search algorithms. This dynamic balance between exploration and exploitation is a well-known strategy for improving convergence in complex landscapes. Further analysis of parameters and convergence is provided in Section 4.3.

# 4 Experiments

## 4.1 Experimental setup

**Models and Quantization.** We conduct experiments on several large language models: Llama 3.1 8B, Qwen 2.5 14B, Qwen 2.5 32B, and Llama 3.3 70B. GPTQ (Frantar et al., 2022) asymmetric quantization is applied to all these models, with a group size of 64 for Llama 3.1 8B and Qwen 2.5 14B, and 128 for Qwen 2.5 32B and Llama 3.3 70B. For calibration, we use 1024 samples from the C4 dataset (Raffel et al., 2019).

**Tasks.** We focus on quantization-aware fine-tuning (QAF) and implement two fine-tuning paradigms: performance-recovery and task-specific. For performance-recovery fine-tuning, we

Table 1: Accuracy (%) of performance-recovery and task-specific. An em dash (–) indicates unobtainable results, as the evaluation script requires strict format alignment with fine-tuning data.

| Method | #Bit | MMLU (5-shot) | | | | | Task-Specific (0-shot) | | |
|---|---|---|---|---|---|---|---|---|---|
| | | Hums. | STEM | Social | Other | Avg. | GSM8K | SQL | ViGGO |
| LLaMA-70B | 16 | *80.30* | *75.26* | *88.94* | *84.58* | *82.23* | – | – | – |
| GPTQ | 4 | 80.09 | 75.04 | 88.79 | 84.39 | 81.81 | – | – | – |
| GPTQ+LoRA | 4+16 | 80.19 | 74.98 | 88.79 | 84.39 | 81.83 | 90.14 | 82.90 | 89.72 |
| QA-LoRA | 4 | 80.11 | 75.01 | 88.89 | 84.29 | 81.81 | 84.32 | 77.65 | 85.35 |
| **LoTA-QAF** | 4 | 80.12 | 75.07 | 88.82 | 84.36 | **81.84** | 87.04 | 78.50 | 86.46 |
| GPTQ | 3 | 77.62 | 71.30 | 86.38 | 82.17 | 79.13 | – | – | – |
| GPTQ+LoRA | 3+16 | 77.58 | 71.27 | 86.48 | 82.23 | 79.14 | 88.17 | 80.20 | 88.09 |
| QA-LoRA | 3 | 77.66 | 71.27 | 86.45 | 82.07 | 79.13 | 82.65 | 75.80 | 76.38 |
| **LoTA-QAF** | 3 | 77.68 | 71.30 | 86.42 | 82.36 | **79.20** | 83.24 | 77.40 | 85.38 |
| GPTQ | 2 | 39.00 | 35.08 | 46.83 | 46.64 | 41.53 | – | – | – |
| GPTQ+LoRA | 2+16 | 52.58 | 45.86 | 65.91 | 62.63 | 56.22 | 67.54 | 72.70 | 79.35 |
| QA-LoRA | 2 | 50.03 | 44.08 | 61.68 | 58.83 | 53.20 | 61.80 | 70.70 | 68.44 |
| **LoTA-QAF** | 2 | 52.96 | 47.10 | 67.34 | 62.83 | **56.98** | 63.98 | 73.40 | 75.38 |
| Qwen-32B | 16 | *77.95* | *82.43* | *88.72* | *84.96* | *82.92* | – | – | – |
| GPTQ | 4 | 77.90 | 82.11 | 88.56 | 83.71 | 82.47 | – | – | – |
| GPTQ+LoRA | 4+16 | 78.13 | 82.46 | 88.59 | 83.52 | 82.59 | 83.09 | 88.70 | 77.29 |
| QA-LoRA | 4 | 77.98 | 82.02 | 88.63 | 83.68 | 82.48 | 76.82 | 86.80 | 70.93 |
| **LoTA-QAF** | 4 | 78.07 | 82.33 | 88.76 | 84.10 | **82.70** | 78.74 | 87.60 | 76.18 |
| GPTQ | 3 | 75.66 | 79.45 | 87.36 | 81.17 | 80.29 | – | – | – |
| GPTQ+LoRA | 3+16 | 75.75 | 79.42 | 87.46 | 81.53 | 80.42 | 82.34 | 88.90 | 74.70 |
| QA-LoRA | 3 | 75.58 | 79.42 | 87.33 | 81.43 | 80.31 | 75.44 | 80.60 | 57.20 |
| **LoTA-QAF** | 3 | 76.00 | 79.51 | 87.59 | 81.43 | **80.53** | 76.04 | 83.20 | 60.94 |
| GPTQ | 2 | 34.03 | 32.54 | 37.60 | 39.46 | 35.68 | – | – | – |
| GPTQ+LoRA | 2+16 | 44.06 | 43.04 | 54.92 | 54.23 | 48.46 | 57.55 | 82.40 | 69.99 |
| QA-LoRA | 2 | 39.34 | 42.02 | 53.04 | 54.49 | 46.30 | 54.74 | 80.40 | 55.44 |
| **LoTA-QAF** | 2 | 47.31 | 45.42 | 60.87 | 59.93 | **52.65** | 57.92 | 82.20 | 63.71 |
| Qwen-14B | 16 | *74.71* | *77.86* | *87.46* | *82.39* | *79.91* | – | – | – |
| GPTQ | 4 | 74.79 | 76.91 | 87.26 | 81.88 | 79.57 | – | – | – |
| GPTQ+LoRA | 4+16 | 74.79 | 77.04 | 87.42 | 81.94 | 79.65 | 80.06 | 87.70 | 74.05 |
| QA-LoRA | 4 | 74.90 | 76.78 | 87.36 | 81.82 | 79.58 | 76.10 | 83.90 | 68.47 |
| **LoTA-QAF** | 4 | 74.86 | 77.20 | 87.52 | 82.14 | **79.77** | 78.37 | 84.50 | 71.10 |
| GPTQ | 3 | 71.07 | 72.41 | 84.24 | 79.79 | 76.19 | – | – | – |
| GPTQ+LoRA | 3+16 | 71.48 | 73.20 | 84.37 | 80.11 | 76.60 | 72.18 | 87.40 | 71.93 |
| QA-LoRA | 3 | 71.16 | 72.79 | 84.21 | 79.88 | 76.31 | 66.49 | 77.30 | 57.36 |
| **LoTA-QAF** | 3 | 71.54 | 72.79 | 84.76 | 80.17 | **76.63** | 70.36 | 79.50 | 64.45 |
| GPTQ | 2 | 30.01 | 32.03 | 33.57 | 32.15 | 31.72 | – | – | – |
| GPTQ+LoRA | 2+16 | 39.17 | 41.29 | 48.13 | 43.90 | 42.66 | 37.23 | 80.60 | 61.59 |
| QA-LoRA | 2 | 33.79 | 36.60 | 42.18 | 49.79 | 39.80 | 34.69 | 58.10 | 43.87 |
| **LoTA-QAF** | 2 | 45.23 | 42.85 | 55.44 | 49.15 | **47.80** | 36.25 | 62.80 | 52.63 |
| LLaMA-8B | 16 | *64.17* | *60.39* | *77.64* | *73.09* | *68.25* | – | – | – |
| GPTQ | 4 | 62.19 | 58.71 | 76.37 | 72.93 | 66.89 | – | – | – |
| GPTQ+LoRA | 4+16 | 62.57 | 58.33 | 76.73 | 73.22 | 67.08 | 70.20 | 76.70 | 88.64 |
| QA-LoRA | 4 | 62.06 | 58.71 | 76.54 | 73.25 | 66.96 | 68.69 | 74.10 | 51.15 |
| **LoTA-QAF** | 4 | 62.95 | 58.55 | 76.70 | 72.93 | **67.18** | 70.05 | 74.60 | 60.30 |
| GPTQ | 3 | 32.48 | 36.66 | 39.42 | 49.08 | 38.61 | – | – | – |
| GPTQ+LoRA | 3+16 | 50.24 | 46.91 | 63.24 | 60.80 | 54.68 | 65.66 | 75.80 | 82.20 |
| QA-LoRA | 3 | 45.55 | 46.94 | 61.33 | 59.35 | 52.37 | 62.17 | 72.20 | 43.86 |
| **LoTA-QAF** | 3 | 49.71 | 49.29 | 65.29 | 62.63 | **55.89** | 63.53 | 73.10 | 55.96 |
| GPTQ | 2 | 25.89 | 27.53 | 26.71 | 26.30 | 26.53 | – | – | – |
| GPTQ+LoRA | 2+16 | 27.21 | 28.64 | 26.91 | 26.68 | 27.35 | 34.12 | 72.50 | 80.70 |
| QA-LoRA | 2 | 25.70 | 26.67 | 27.72 | 26.62 | 26.56 | 17.36 | 56.20 | 37.18 |
| **LoTA-QAF** | 2 | 28.25 | 26.90 | 28.44 | 30.51 | **28.49** | 19.48 | 67.00 | 41.98 |

utilize the Alpaca (Taori et al., 2023) and subsequently evaluate 5-shot performance on the Massively Multitask Language Understanding (MMLU) benchmark (Hendrycks et al., 2020). For task-specific fine-tuning, we select three datasets: GSM8K (Cobbe et al., 2021), with 7.47k training and 1.32k test samples; SQL generation (Yu et al., 2018; Zhong et al., 2017), with 30k training and 1k test samples; ViGGO (Juraska et al., 2019), with 5.1k training and 1.08k test samples. We use the lm-eval framework (Gao et al., 2024) to test the MMLU benchmark and follow the custom evaluation framework from HALO (Ashkboos et al., 2025) for the task-specific evaluations. These datasets are chosen because they represent diverse specialized tasks that present unique challenges and require distinct model capabilities. Their inclusion allows us to clearly illustrate the practical significance of the QAF approaches in developing quantized models for specific applications.

**Baselines and Hyper-parameters.** We benchmark LoTA-QAF against two state-of-the-art methods: LoRA (Hu et al., 2022) and QA-LoRA (Xu et al., 2023). For fine-tuning quantized models to recover performance and align them with specific tasks, LoRA remains a widely adopted method (applied to GPTQ-quantized models, it can be considered a QLoRA (Dettmers et al., 2023) variant). Although LoRA introduces 16-bit adapters, resulting in reduced inference efficiency, the 16-bit precision of its adapters also enables more precise adjustment of the quantized weights. Moreover, as the closest work to ours, QA-LoRA maintains low-bit inference efficiency by lossless merging the adapters into zero factors.

Following QA-LoRA, we use a paged AdamW optimizer, a maximum gradient norm of $0.3$, a batch size of $64$, a source length of $1024$, and a target length of $256$. For performance-recovery fine-tuning, we set the learning rate to $1 \times 10^{-5}$ for the 8B and 14B models, and $5 \times 10^{-6}$ for the 32B and 70B models. The number of fine-tuning steps is $300$ for Alpaca. For task-specific fine-tuning, the learning rate is set to $5 \times 10^{-4}$ for the 8B and 14B models, and $1 \times 10^{-4}$ for the 32B and 70B models. Single-epoch experiments are performed on the training sets of GSM8k, SQL generation, and ViGGO. Regarding adapter settings, the rank $r$ is $64$ for the 8B and 14B models, and $32$ for the 32B and 70B models. Additionally, the coefficient $\alpha$ is twice the rank. Regarding the hyper-parameters of LoTA-QAF, we set the threshold $\omega$ to $0.75r$ for Alpaca, GSM8K and SQL generation, and $\omega$ to $0.875r$ for ViGGO. The dynamic percentile-based threshold, $\sigma_t$, is initialized to the top 5% and linearly decays to 0.1% during the first 80% of the training phase. For the final 20% of training (i.e., from 80% to 100% completion), it is fixed at 0.01%. Section 4.3 provides a detailed analysis of $\omega$ and $\sigma_t$. All experiments are conducted on one NVIDIA A800 GPU.

## 4.2 Main Results

**Performance-Recovery Fine-Tuning.** For performance-recovery fine-tuning, our objective is to restore the performance of quantized models to the level of the 16-bit base models. As Table 1 shows, across four model sizes and three quantization bits, the three quantization-aware fine-tuning (QAF) methods all achieve improvements compared to the quantized baseline. For instance, with 3-bit quantization, the fine-tuned performance surpasses that of the quantized model by up to 17.28% on Llama 3.1 8B. Similarly, for 2-bit quantization, the fine-tuned performance exceeds that of the quantized model by up to 16.97% on Qwen 2.5 32B. This indicates that in the low-bit regimes, there remains a performance potential that existing Post-Training Quantization (PTQ) methods may not fully exploit, but which can be harnessed through fine-tuning. This highlights the significant development potential of QAF methods.

Furthermore, LoTA-QAF outperforms LoRA (GPTQ+LoRA in Table 1) and QA-LoRA. On Qwen 2.5 14B 2-bit, LoTA-QAF improves performance by 5.14% compared to LoRA, the second-best performing method in this comparison. This superiority is attributed to LoTA-QAF directly adjusting the quantized weights and aligning with the quantization grid, which effectively recovers performance lost by quantization. In contrast, while LoRA utilizes a 16-bit adapter, it lacks a direct quantization-aware mechanism. As noted earlier, QA-LoRA only adjusts the zero factors in group-wise quantization, which limits its fine-tuning capability.

**Task-Specific Fine-Tuning.** For task-specific fine-tuning, the objective is to enable quantized models to acquire the knowledge and patterns specific to particular tasks. This necessitates that the model learn fine-grained capabilities specific to the task, rather than merely recovering general abilities. As shown in Table 1, LoTA-QAF outperforms QA-LoRA but is surpassed by LoRA. LoRA's 16-bit adapter possesses a higher representational capacity, enabling it to more effectively capture

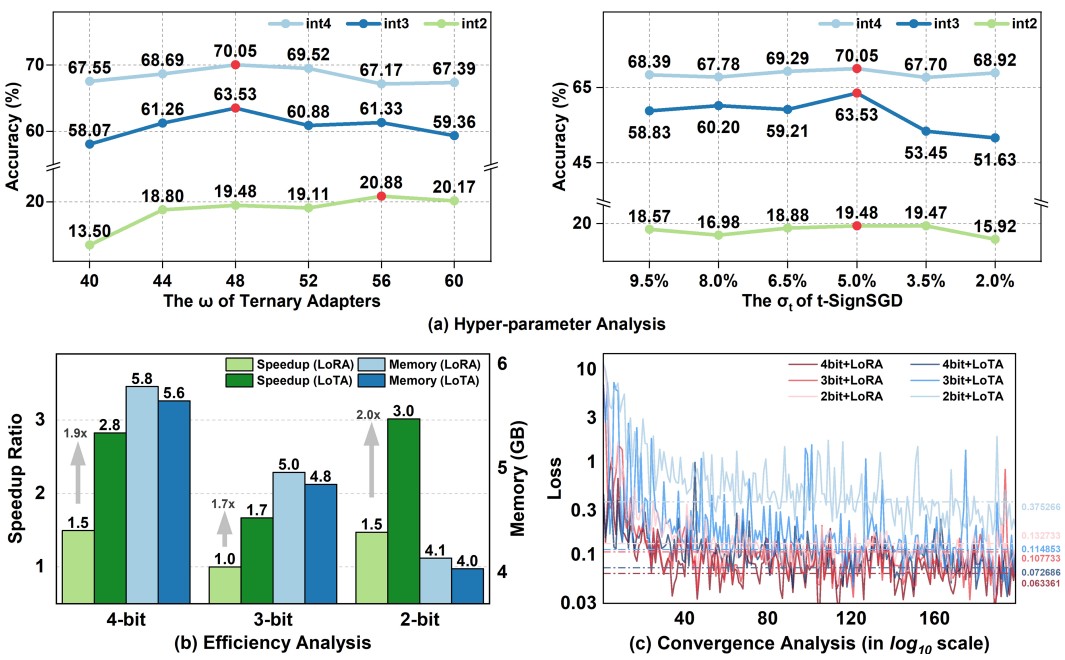

Figure 4: Further analysis of LoTA-QAF

these complex task-specific details. This higher precision is particularly crucial for heavily quantized models (e.g., 2-bit models). However, it is important to note the difference in inference efficiency: LoTA-QAF and QA-LoRA perform low-bit inference once their adapters are merged, whereas inference with LoRA requires computation involving the 16-bit adapter (as detailed in Section 4.3).

## 4.3 Further Analysis of LoTA-QAF

**Hyper-parameter Analysis.** We investigate the hyper-parameters $\omega$ of the ternary adapter (TA) and $\sigma_t$ of ternary signed gradient descent (t-SignSGD) through parameter experiments conducted on GSM8K with 4, 3 and 2 bit-widths of the Llama 3.1 8B model, with results shown in Fig. 4. First, TA employs a threshold $\omega$ to determine the extent to which it influences the quantized weights. This threshold $\omega$ is analogous to the LoRA coefficient $\alpha$ in Equation 1. A smaller $\omega$ allows values in the auxiliary matrix $\Delta \mathbf{W}$ to affect the quantized weights more readily, as detailed in Equations 3 and 5. Specifically, the threshold $\omega$ is a hyper-parameter related to the rank $r$. We set the rank $r$ to 64 for the 8B model and experiment with $\omega$ values of $\{40, 44, 48, 52, 56, 60\}$. We observe that setting $\omega$ to 48 yields favorable performance across all three bit-widths. Notably, for 2-bit quantization, a relatively larger $\omega$ achieves better experimental results. This is because, in 2-bit quantization, each quantized weight possesses only four possible values. Consequently, updating the quantized weights with a larger $\omega$, which exerts a more conservative influence, leads to improved outcomes.

Second, the dynamic percentile-based threshold $\sigma_t$ (where $t$ is the iteration step) acts as a learning rate in t-SignSGD, as explained in Equation 6. We conduct parameter experiments by selecting initial $\sigma_t$ values corresponding to the top $\{9.5\%, 8.0\%, 6.5\%, 5.0\%, 3.5\%, 2.0\%\}$ of gradient magnitudes. For the 8B model, an initial $\sigma_t$ of 5.0% (representing the top 5.0% of gradients) consistently yields favorable results. Conversely, selecting smaller initial $\sigma_t$ values leads to a noticeable decline in performance for both 3-bit and 2-bit quantization. This observation can be attributed to the nature of task-specific fine-tuning, which necessitates more substantial adjustments to model parameters compared to recovery performance. This aligns with our general learning rate strategy for LoRA and QA-LoRA, where the learning rate for recovery performance is set lower than that for task-specific fine-tuning. Further hyper-parameter analysis is provided in the Appendix B.

**Efficiency Analysis.** We measure the inference speed and memory usage of LoRA (utilizing a forward pass with 4-bit quantized weights and 16-bit adapters) and LoTA (where the adapter is losslessly merged into the quantized weights). These measurements are conducted on the ViGGO dataset using the Llama 3.1 8B model with 4, 3 and 2 bit-widths quantization. For 4-bit and 2-bit quantized models, the TritonV2QuantLinear kernel is used. For the 3-bit quantized model, the

TorchQuantLinear kernel is employed. To evaluate inference speed, we measure throughput (tokens per second) with batch sizes ranging from 8 to 128 and a maximum inference length of 512 tokens. The speedup ratio, benchmarked against the slowest configuration (3-bit quantized weights with LoRA), is reported in Fig.4. LoTA achieves speedups of 1.9x, 1.7x, and 2.0x over LoRA across these respective bit-widths, revealing the efficiency gains from merging the adapter post fine-tuning. In terms of memory usage, LoTA also exhibits a slight advantage.

**Convergence Analysis.** Regarding convergence, we evaluate LoRA and LoTA on Llama 3.1 8B for SQL generation with 4, 3 and 2 bit-widths quantization. LoRA achieves best convergence across all three bit-widths. Notably, at 2-bit, LoRA reaches a convergence loss of $0.132$, whereas LoTA achieves $0.375$, showing LoRA's 16-bit adapter stability. However, LoTA, optimized with t-SignSGD, also converges. For 4-bit and 3-bit quantization, its convergence loss differs from LoRA $< 0.01$. Considering that LoTA can be losslessly merged into the quantized weights, whereas LoRA cannot, this makes LoTA a highly competitive method, especially when deployment efficiency is a priority.

## 5    Conclusion

Fine-tuning quantized LLMs for edge devices suffers from merging issues. LoTA-QAF enables in-grid adjustment of all quantized weights and lossless merging of ternary adapters using novel ternary adaptation and ternary signed gradient descent (t-SignSGD), and preserves performance by avoiding merging-induced accuracy loss. The Appendix details ternary adapters implementation, further parameters analysis, training efficiency analysis, limitations, and future work.

## 6    Acknowledgments

The work of Long Shi was supported by the National Natural Science Foundation of China under Grant 62201475 and Sichuan Science and Technology Program under Grant 2024NSFSC1436. The work of Xuming Hu was supported by the National Natural Science Foundation of China (Grant No.62506318); Guangdong Provincial Department of Education Project (Grant No.2024KQNCX028); CAAI-Ant Group Research Fund; Scientific Research Projects for the Higher-educational Institutions (Grant No.2024312096), Education Bureau of Guangzhou Municipality; Guangzhou-HKUST(GZ) Joint Funding Program (Grant No.2025A03J3957), Education Bureau of Guangzhou Municipality.

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

# A  Implementation of Ternary Adapters

The PyTorch framework does not natively support `ternary` or `torch.int2` data formats. Furthermore, the PyTorch deep learning training ecosystem primarily relies on floating-point computations. Therefore, to seamlessly integrate into this established system, we simulate `ternary` data types by representing values of $\{-1, 0, 1\}$ using `bfloat16`.

For the process of forming the ternary matrix $\hat{\mathbf{W}}$ via the auxiliary matrix $\Delta \mathbf{W} = \mathbf{A}_\mathrm{T} \mathbf{B}_\mathrm{T}$ (as described in Eq. (3) of the main text), we implement custom operations using `Triton` to enhance the execution efficiency on GPUs. Moreover, we update the quantized weights $\mathbf{W}_\mathrm{int}$ which requires boundary checks [1] [2]. Specifically, through kernel fusion, `Triton` combines the formation of $\hat{\mathbf{W}}$ and the application of boundary checks (boolean masks) into a single, optimized GPU kernel. This fusion minimizes overhead and contributes to maintaining acceptable training costs for the LoTA-QAF.

Notably, the implementation of ternary adapters faces challenges primarily due to framework limitations. Methodologically, LoTA-QAF employs ternary adapters, which offer significant advantages in storage and computational efficiency over 16-bit adapters. Consequently, further engineering for production-level deployment is valuable and represents a promising avenue for future work (see Section F). Currently, the training costs are acceptable with an implementation based on PyTorch and accelerated by `Triton`. A detailed discussion on training efficiency is available in Section C.

# B  Hyper-parameters of LoTA-QAF

To further analyze the hyperparameters of LoTA-QAF, we focus on two aspects: i) the Llama 3.1 8B model across various datasets, specifically (a) MMLU, (b) SQL, and (c) ViGGO (with GSM8K detailed in the main text); and ii) the Qwen 2.5 14B and 32B models on the GSM8K dataset, as shown in (d) and (e). For all analyses of $\omega$ (defined in Eq. (3)), we set $\sigma_t$ to 5%. For the analysis of $\sigma_t$, we set $\omega$ to $0.75r$ (except $0.875r$ for ViGGO), where $r$ is the rank. Other settings are detailed in Section 4.1.

Overall, more detailed hyperparameter tuning can reveal accuracies (e.g., as shown in Fig. 5 (b)) that exceed those reported in main text Table 1. This is because different quantization bit-widths, datasets, and model sizes often have their own more suitable parameter settings. The objective of our further hyperparameter analysis is to identify the characteristics of these settings and understand the LoTA-QAF properties they reflect.

For the Llama 3.1 8B model on various datasets: i) the $\omega$ parameter of the ternary adapters (TA, defined in Eq. (3)) primarily determines how intensity $\hat{\mathbf{W}}$ adjusts the quantized weights. For 4-bit quantized models, a smaller $\omega$ can potentially yield gains (e.g., Fig. 5 (b), int4), attributed to the robustness and adjustability of 4-bit models. However, for 3-bit and 2-bit quantization, a smaller $\omega$ generally leads to poorer performance (e.g., Fig. 5 (b) and (c), int3 and int2). Indeed, for 2-bit quantization, setting a higher $\omega$ to limit the adjustment intensity of $\hat{\mathbf{W}}$ on quantized weights conversely achieves higher accuracy (e.g., Fig. 5 (c), int2). ii) the $\sigma_t$ parameter of t-SignSGD (defined in Eq. (6)), which functions as the learning rate in t-SignSGD, selects for update ternary adapter weights with gradient magnitudes in the top $x\%$. For both SQL and ViGGO datasets (Fig. 5 (b) and (c)), employing a larger $\sigma_t$ results in higher accuracy. This aligns with our LoRA training configurations. Specifically,

---

[1] Two main approaches exist for performing these boundary checks: one is to decode the boundaries from the quantized weights during each forward propagation, which adversely affects training efficiency. Consequently, we adopt an alternative approach: we first identify the boundaries and then store them using a boolean packing technique (`def pack_bool_tensor()`) for use in the forward propagation. While this method introduces some memory overhead, the cost is acceptable as each boolean flag (1 bit) is packed.

[2] Boundary checks are not strictly necessary during the training phase for 4-bit or even 3-bit quantization, as the probability of encountering boundary conditions is relatively low in schemes that utilize 16 or 8 distinct values, respectively, especially considering that the number of values at the boundaries is typically smaller than that of values near the center. However, for 2-bit quantization, which involves only 4 distinct values, boundary checks become crucial. Specifically, neglecting boundary checks during training and only applying them when loading or merging the adapter can lead to inconsistencies between training behavior and the operational results of the adapter for 2-bit quantized models. In our experiments, we enable boundary checks universally (including in training efficiency tests) to ensure consistency, even though they could be optionally disabled for 4-bit and 3-bit scenarios during training without a significant adverse impact.

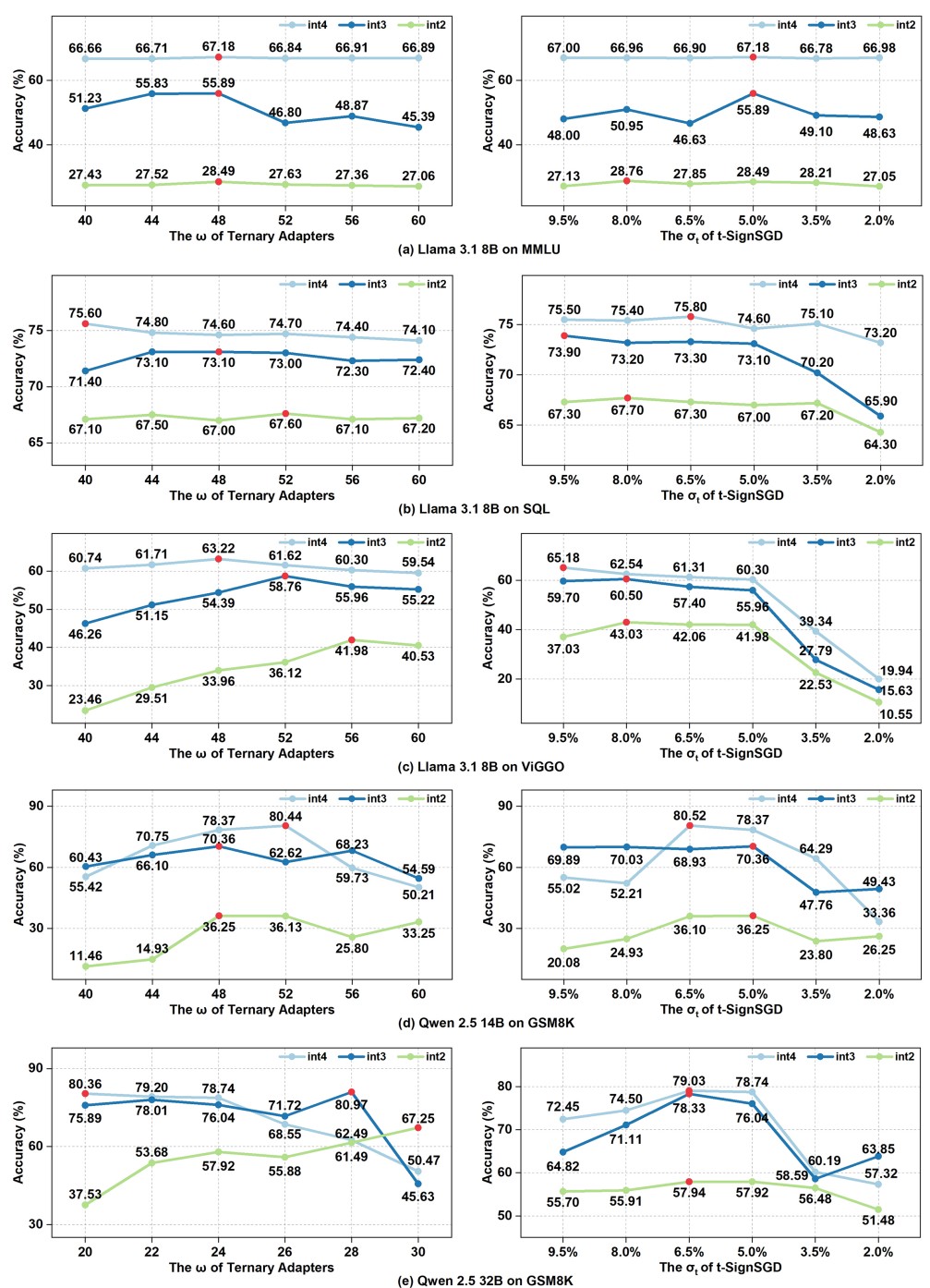

Figure 5: Parameter analysis for Llama 3.1 8B on various datasets in (a) MMLU, (b) SQL and (c) ViGGO. Parameter analysis for Qwen 2.5 14B and 32B on the GSM8K dataset in (d) and (e).

task-specific fine-tuning requires a higher learning rate than performance-recovery fine-tuning, as learning specific tasks demands greater model weight adjustments than performance recovery.

For the Qwen 2.5 14B and 32B models on the GSM8K dataset: i) setting $\omega$ to $0.9375r$ leads to a significant decrease in accuracy for 4-bit and 3-bit quantization (e.g., Fig. 5 (d) and (e)). Conversely, for 2-bit quantized weights, this $\omega$ ($0.9375r$) value achieves higher accuracy. This is consistent with our earlier findings. ii) setting $\sigma_t$ to either $6.5\%$ or $5.0\%$ is a robust choice. An overly small $\sigma_t$ (e.g.,

2.0%) results in poorer accuracy (e.g. Fig. 5 (d) and (e) int4 and int2), limiting its ability to learn effectively. Furthermore, due to the high cost of extensive hyperparameter testing for 70B models, we offer a baseline configuration: setting $\omega$ to $0.75r$ and $\sigma_t$ to 5.0% generally yields good accuracy. For further improvement, we suggest trying $\sigma_t = 2.0\%$ for performance-recovery fine-tuning and $\sigma_t = 6.5\%$ for task-specific fine-tuning. Regarding the $\omega$ setting for 70B models, suggests that experimenting with lower $\omega$ values may improve accuracy, which is attributed to the 4-bit quantized 70B model's robustness.

## C    Training Efficiency Analysis

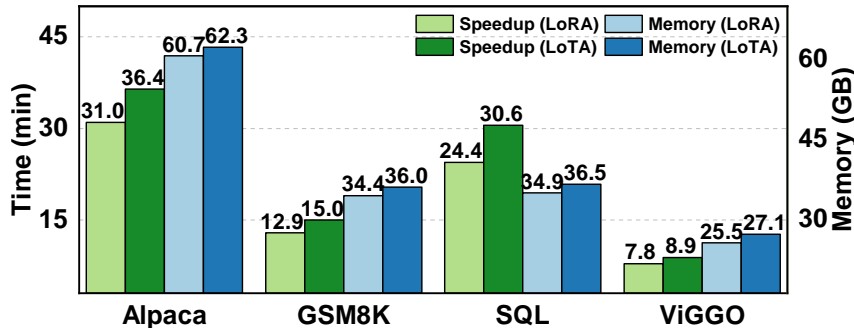

Figure 6: Comparative analysis of training efficiency for LoRA versus LoTA. The evaluation utilized the Llama 3.1 8B model with 4-bit precision on four datasets. Metrics reported are total training execution time and peak memory usage. For parameter details, including a batch size of 64, and other settings, please refer to the Section 4.1.

Our simulated `ternary` adapter utilizes `bfloat16` for implementation due to the lack of native support for the `ternary` dtype within the PyTorch framework. Consequently, LoTA-QAF does not exhibit a training efficiency advantage over LoRA. Specifically, LoTA requires 14.1%-25.4% more training time compared to LoRA. This increased training time is primarily attributed to the implementation of Eq. (4) and the process of aligning with the dequantization process of GPTQModel. Despite this higher training cost, LoTA's ternary adapters can be losslessly merged into quantized weights, a capability that LoRA does not offer. In terms of memory, LoTA incurs an additional 2.6%-6.3% overhead compared to LoRA. This overhead derives from the boundary checks detailed in Section A, though the overall cost remains modest. Section F discusses feasible directions for further enhancing LoTA's training efficiency.

## D    Ternary Adapter Weight Changes under t-SignSGD Training

Table 2: Analysis of ternary adapter weight changes.

| Metric | Value | Percentage | Description |
|---|---|---|---|
| Total Parameters | 167,772,160 | 100% | Total adapter parameters across 32 layers. Adapters are applied to 7 key projections within the attention and MLP blocks of each layer. |
| Increased Parameters | 28,952,566 | 17.26% | Count of weights that changed in the positive direction. |
| Decreased Parameters | 28,960,070 | 17.26% | Count of weights that changed in the negative direction. |
| Unchanged Parameters | 109,859,524 | 65.48% | Count of weights that are unchanged. |

To better understand the dynamics of our optimizer, we analyzed the parameter changes of the ternary adapters for Llama 3.1 8B after fine-tuning on the Alpaca dataset. Table 2 shows that the

rate of change in the adapters is nearly identical in both positive and negative directions (17.26% vs. 17.27%). This suggests that the adaptation is not a simple unidirectional shift but a balanced, bidirectional fine-tuning process to fit the target task. Moreover, one-third of the parameters changed during adaptation, while two-thirds remained unchanged. This phenomenon is a result of the t-SignSGD design, specifically the adaptive threshold $\mathbb{I}_{|g_t|>\max(\tau,\sigma_t)}$. This mechanism ensures our updates are based on a large gradient magnitude, which implies a high probability that the sign is correct. Conversely, it filters out updates associated with small gradient signals, which can be noisy or ill-scaled.

# E Limitations

Regarding training efficiency, LoTA-QAF keeps its training costs at a level comparable to LoRA through `Triton`, but LoTA-QAF has not fully realized the potential high efficiency inherent in ternary adapters. A common challenge in current low-bit research is bridging the gap between theoretical and realized efficiency. Although `Triton` and `CUTLASS` offer pathways to achieve efficient implementations, its implementation is challenging.

Regarding convergence, LoTA-QAF exhibits less favorable convergence compared to LoRA (discussed in Section 4.3). This can be attributed to two main aspects: firstly, ternary adapters adjust the quantized weights based on the quantization grid. For 4-bit and 3-bit quantization, which involve 16 and 8 possible values respectively, such adjustments demonstrate reasonable robustness. However, under 2-bit quantization with only four possible values, these adjustments exhibit higher volatility. Secondly, t-SignSGD is a novel update mechanism that operates without a learning rate, and we do not incorporate first or second-order momentum. Moreover, the primary constraint is a linear decay of the threshold $\sigma_t$ (as defined in Eq. (6)). While the t-SignSGD approach is effective for updating ternary adapters, our exploration of its optimization remains limited.

# F Future Work

We believe that the current training efficiency disadvantage of the ternary adapter (TA) can be addressed. Although TA involves more computational logic during forward propagation, the operational advantages of `ternary` arithmetic can be further exploited. The auxiliary matrix $\Delta\mathbf{W} = \mathbf{A}_T\mathbf{B}_T$ is calculated using `bfloat16`, but executing operations on ternary values $\{-1, 0, 1\}$ could further improve efficiency. The forward execution logic in Eq. (5) is merged with the dequantization process of GPTQModel. However, given that $\hat{\mathbf{W}}$ is a sparse ternary matrix, the potential for dedicated sparse computation logic warrants exploration. Such implementations, however, must consider not only computational efficiency but also ensure the correct functioning of the gradient mechanism for the `ternary` data type within the PyTorch framework.

Our exploration of t-SignSGD has been limited. Further research could focus on incorporating momentum mechanisms, refining constraints on t-SignSGD, and applying optimization techniques (e.g., cosine annealing, cyclical learning rates, or adaptive step-size strategies). In t-SignSGD, we replaced the learning rate with a dynamic percentile-based threshold, $\sigma_t$, to select the top-$x$% of gradient signs for directly updating the ternary adapter weights within the set $\{-1, 0, 1\}$. Under this mechanism, a more detailed analysis of gradient distribution characteristics could be pursued to guide the selection of ternary adapter weights for updates, potentially offering improved convergence properties.

Further enhancements to LoTA-QAF could also be explored by expanding the representational range of the auxiliary matrix $\hat{\mathbf{W}}$. Currently, this matrix is constrained to the set $\{-1, 0, 1\}$. However, the LoTA mechanism, as outlined in Eq. (3), could be extended to broaden the value range of $\hat{\mathbf{W}}$ to include, for instance, $\{-2, -1, 0, 1, 2\}$, or even a wider set of integers. Such an extension could further enhance the optimization capability for quantized weights. It is important to note, though, that this approach might be less suitable for `int2` or even `int3` quantization formats due to the potentially large adjustment step sizes. Nevertheless, for `int4` or `int8` quantization, exploring this direction is highly valuable.

