# OpenReview forum: "LoTA-QAF: Lossless Ternary Adaptation for Quantization-Aware Fine-Tuning"
_NeurIPS.cc/2025/Conference — NeurIPS 2025 poster_

### Official Review · Reviewer_Qp8Z · 2025-06-30

**Clarity:** 3
**Significance:** 3
**Originality:** 3
**Rating:** 4
**Confidence:** 4

**Summary:**

This paper introduces a novel adaptation method for quantization-aware fine-tuning (QAF), enabling lossless merging of adaptation weights into quantized weights. The method includes three main components: (1) custom-designed ternary adaptation that adjusts quantized weights within the quantization grid, (2) a lossless merging mechanism using auxiliary and offset matrices, and (3) ternary signed gradient descent (t-SignSGD) for optimization.

**Questions:**

please see the weeknesses.

**Ethical Concerns:**

["NO or VERY MINOR ethics concerns only"]

**Final Justification:**

Thanks to the author for the Rebuttal. I have read it carefully, which further strengthens the contributions. I will keep my positive score.

**Limitations:**

yes

**Quality:**

3

**Strengths And Weaknesses:**

[Strengths]
1. The paper introduces an innovative QAF method with elegantly designed ternary adaptation that enhances learning capability by enabling direct adjustment of quantized weights rather than merely modifying quantization parameters.
2. The ternary constraint ensures that weight adjustments remain within the quantization grid, completely avoiding the additional quantization errors introduced during adapter merging.
3. The design of the t-SignSGD optimizer with dynamic percentile-based thresholding is remarkably ingenious, utilizing minimal hyperparameters while effectively solving the discrete optimization problem.
4. The LoTA-QAF works effectively across different quantization bit-widths (2-bit, 3-bit, 4-bit) without requiring method-specific modifications. This flexibility makes it a versatile solution that can adapt to various deployment constraints and hardware capabilities.

[Weaknesses]
1. Compared to LoRA, LoTA-QAF seems restrictive in learning complex task-specific patterns, particularly in low-bit settings. How significant is the representational capacity gap between ternary adapters and full-precision adapters? It’s necessary to provide some insights.
2.  For t-SignSGD's convergence properties, the authors provide only relatively brief descriptions and lack more in-depth discussion or theoretical analysis, specifically regarding convergence behavior in discrete optimization spaces.
3. The current baseline evaluation is constrained, with comparisons limited to QA-LoRA and LoRA, while other QAF methods remain unexamined.

---

> ### Author Rebuttal · Authors · 2025-07-30
>
> We thank the reviewer for recognizing the innovation and practical advantages of LoTA-QAF. We will address your insightful questions to further clarify the technical details.
>
> > **Q1**: Compared to LoRA, LoTA-QAF seems restrictive in learning complex task-specific patterns, particularly in low-bit settings. How significant is the representational capacity gap between ternary adapters and full-precision adapters? It’s necessary to provide some insights.
>
> > **A1**: Let's further analyze the performance on task-specific fine-tuning. Across the four model sizes we evaluated (8B, 14B, 32B, and 70B), the performance gap between LoTA-QAF and 16-bit LoRA is smallest on the 70B model. However, the performance of LoTA on the 2-bit quantized 8B model is significantly lower than that of LoRA.
> >
> > This observation aligns with the design philosophy of LoTA-QAF. For large models like the 70B version, the representational capacity is sufficient for LoTA's coarse-grained +/−1 adjustments on the discrete quantization grid to effectively align the model. Furthermore, while LoTA-QAF always applies a ternary influence on the quantized weights, the effect of these adjustments is finer-grained on 4-bit models than on 2-bit ones. A 4-bit weight has 16 possible values whereas a 2-bit weight has only 4. This means each +/−1 adjustment on the 4-bit grid is relatively more fine-grained. This lower precision on the 2-bit grid limits the adapter's ability to make the subtle changes required for fine-tuning, which explains its lower performance.
>
> ---
>
> > **Q2**: For t-SignSGD's convergence properties, the authors provide only relatively brief descriptions and lack more in-depth discussion or theoretical analysis, specifically regarding convergence behavior in discrete optimization spaces.
>
> > **A2**: We sincerely thanks for your positive feedback on t-SignSGD and your insightful questions regarding its convergence properties. Our proposed t-SignSGD can be understood as a gradient-guided discrete search algorithm whose design includes several mechanisms to ensure convergence.
> >
> > We will elaborate on this from three perspectives: 1) The Challenge of Formal Convergence Analysis, 2) Theoretical Grounding in SignSGD, and 3) Analysis of the t-SignSGD Update Rule for Convergence.
> >
> > **1. The Challenge of Formal Convergence Analysis**
> >
> > A formal convergence proof for our method is highly complex. Firstly, t-SignSGD operates on a discrete and bounded parameter space of {-1, 0, 1}. Updates are discrete jumps (e.g., from 0 to 1) rather than infinitesimal steps in a continuous space, rendering standard gradient-based convergence analyses inapplicable.
> >
> > Secondly, the forward pass from the ternary adapters to the final loss involves non-differentiable operations, notably the indicator function $\mathbb{I}\_{|\Delta \mathbf{W}\_{ij}| > \omega}$ used to form the ternary matrix $\hat{W}$. To enable backpropagation, we necessarily employ the Straight-Through Estimator (STE). The resulting "pseudo-gradient" is an effective proxy but does not perfectly reflect the true gradient of the loss with respect to the adapter weights.
> >
> > Finally, even for standard SignSGD on continuous weights, theoretical analysis is a complex field, relying on specific geometric assumptions. Therefore, the following discussion focuses on providing a clear theoretical grounding and a mechanistic explanation.
> >
> > **2. Theoretical Grounding in SignSGD**
> >
> > The foundation of t-SignSGD lies in SignSGD. The core insight from SignSGD literature is that the sign of the gradient is often a more robust descent direction than its magnitude, which can be noisy or ill-scaled. The reliability of the sign is closely tied to the signal-to-noise ratio. A large gradient magnitude implies a high probability that the sign is correct.
> >
> > The t-SignSGD adapts this principle to drive state transitions within the discrete {-1, 0, 1} space. The optimization of ternary adapters is a combinatorial problem with an immense search space of $3\^{D\_{in×r}}$ possibilities. Since brute-force search is intractable, the STE-computed gradient acts as a high-quality heuristic, transforming this intractable search into a manageable, iterative optimization process by indicating the most promising update directions.
> >
> > **3. Analysis of the t-SignSGD Update Rule for Convergence**
> >
> > The convergence behavior of t-SignSGD is primarily governed by its update rule, which is designed for the discrete and bounded nature of the ternary adapters.
> > The key component is the indicator function $\mathbb{I}_{|g_t| > \max(\tau, \sigma_t)}$ in Eq. (6), which acts as an adaptive and dynamic selection mechanism. It ensures stable convergence in two ways:
> >
> > **a) Stability via Noise Filtering (Preventing Oscillation)**: In discrete optimization, small, noisy gradients can cause parameters to oscillate unstably between states (e.g., flipping between 0 and 1). Our thresholding mechanism filters out these low-magnitude gradients, ensuring that updates are only performed when the gradient signal is strong and reliable. This addresses the core challenge in sign-based methods related to low signal-to-noise ratios, effectively preventing unstable "jitter" and promoting a smoother convergence path.
> >
> > **b) Convergence via Annealing-like Dynamics**: The decaying percentile threshold $\sigma_t$, introduces an annealing-like, coarse-to-fine search strategy. *Early in training ($\sigma_t$ is high)*, only the top percentile of gradients triggers updates. This focuses the optimization on the most impactful parameters, allowing the model to quickly perform "broad-stroke" adjustments and find a promising region in the vast solution space. *Later in training ($\sigma_t$ is low)*, the threshold decreases, allowing for more fine-grained adjustments. This enables the model to refine its solution within the promising region, analogous to the exploitation phase in classic search algorithms. This dynamic balance between exploration and exploitation is a well-known strategy for improving convergence in complex landscapes.
> >
> > In summary, while our current implementation of t-SignSGD is a concise first-order method, its design is deeply principled. It robustly guides a discrete search by leveraging an adaptive thresholding mechanism that ensures stability and promotes convergence through an effective annealing-like dynamic.
>
> ---
>
> > **Q3**: The current baseline evaluation is constrained, with comparisons limited to QA-LoRA and LoRA, while other QAF methods remain unexamined.
>
> > **A3**: Let's consider another QAF method, IntLoRA. It is designed for efficient merging, as noted by its authors: “For inference, IntLoRA weights can be naturally merged into quantized pre-trained weights through efficient integer multiplication or bit-shifting, eliminating additional post-training quantization.” [1]
> >
> > In our view, IntLoRA employs an innovative mechanism to quantize the adapter more finely, which allows for a better merge within quantized weights. The test results on LLaMA 3.1 8B are as follows:
> >
> > | #Bit | Method | MMLU-Avg. | GSM8k | SQL | ViGGO |
> > | :--- | :--- | :---: | :---: | :---: | :---: |
> > | **4-bit** | QA-LoRA | 66.96 | 68.69 | 74.10 | 51.15 |
> > | | IntLoRA | 66.35 | 68.81 | 74.18 | 54.73 |
> > | | LoTA-QAF | 67.18 | 70.05 | 74.60 | 60.30 |
> > | **3-bit** | QA-LoRA | 52.37 | 62.17 | 72.20 | 43.86 |
> > | | IntLoRA | 51.09 | 61.60 | 69.43 | 40.22 |
> > | | LoTA-QAF | 55.89 | 63.53 | 73.10 | 55.96 |
> > | **2-bit** | QA-LoRA | 26.56 | 17.36 | 56.20 | 37.18 |
> > | | IntLoRA | 26.74 | 15.31 | 51.29 | 33.49 |
> > | | LoTA-QAF | 28.49 | 19.48 | 67.00 | 41.98 |
> >
> > At 4-bit precision, IntLoRA outperforms QA-LoRA but falls short of LoTA-QAF. However, at 2-bit and 3-bit precision, IntLoRA's performance is significantly worse. We hypothesize that this is because the IntLoRA method focuses solely on adjusting the quantized weights without modifying the quantization parameters.
> >
> > *[1] Hang Guo, Yawei Li, Tao Dai, Shu-Tao Xia, and Luca Benini. "IntLoRA: Integral Low-rank Adaptation of Quantized Diffusion Models." Forty-second International Conference on Machine Learning, 2025.*
> ---

---

> > ### Comment · Reviewer_Qp8Z · 2025-08-06
> >
> > Thanks to the author for the Rebuttal. I have read it carefully, which further strengthens the contributions. I will keep my positive score.

---

> > > ### Author Response · Authors · 2025-08-06
> > >
> > > We sincerely thank you for your detailed review and positive feedback. We are pleased that our rebuttal helped clarify the contributions of our work. We will elaborate on the t-SignSGD discussion in the final manuscript to clearly present our contributions. Thank you again for your recognition and support.
> > >
> > > Best regards.

---

### Official Review · Reviewer_abLM · 2025-07-02

**Clarity:** 2
**Significance:** 3
**Originality:** 2
**Rating:** 4
**Confidence:** 3

**Summary:**

This paper proposes LOTA-QAF, a quantized PEFT and loseless merging method for Ternary Quantization. Specifically, the proposed method performs lossless merging by limiting the truncation range of the ternary adapter and adding it to the weights, while the offset part is added to the quantization offset through summation. The proposed method showed comparable performance to QAlora and even better performance at 2bit.

**Questions:**

I recommend that the authors add a more detailed explanation of why this is a lossless merging technique. If I understand correctly, QALoRA achieves this by sacrificing granularity. However, this paper seems to argue that lossless merging is possible without any such cost.

**Ethical Concerns:**

["NO or VERY MINOR ethics concerns only"]

**Final Justification:**

My initial concern was that LOTA-QAF seemed to claim that lossless merging was possible at no cost. However, the authors have clarified that this is achieved by sacrificing accuracy through ternary operations. They also clearly distinguished between the proposed t-signSGD and signSGD. As most of my concerns have now been resolved, I will keep my score at Borderline Accept.

**Limitations:**

yes

**Quality:**

3

**Strengths And Weaknesses:**

Strength
- The paper is well-written and easy to follow.
- The proposed LOTA-QAF and t-signSGD are simple yet demonstrate high performance.
- Experiments were also conducted on large-scale models, such as Llama-70B.

Weakness
- It is difficult to understand why the proposed method is considered lossless merging. Based on the provided figure (Fig. 3), it seems that when per-tensor quantization is performed, the process is not lossless due to the information loss that occurs when the offset weights are merged into a single scalar offset factor.
- The proposed t-signSGD does not appear to be significantly different from the previously proposed SignSGD.
- Limited evaluation: Evaluations on recent LLMs, such as Llama3, or on commonsense QA tasks were not conducted.

---

> ### Author Rebuttal · Authors · 2025-07-29
>
> We thank the reviewer for recognizing the effectiveness of our method. We will address your insightful questions to further clarify the technical details.
>
> > **Q1**: *1.1* It is difficult to understand why the proposed method is considered lossless merging. Based on the provided figure (Fig. 3), it seems that when per-tensor quantization is performed, the process is not lossless due to the information loss that occurs when the offset weights are merged into a single scalar offset factor.
> >
> > *1.2* I recommend that the authors add a more detailed explanation of why this is a lossless merging technique. If I understand correctly, QALoRA achieves this by sacrificing granularity. However, this paper seems to argue that lossless merging is possible without any such cost.
>
> > **A1**: This question touches upon a crucial aspect of our method, and we appreciate the opportunity to clarify. In our work, the term "lossless merge" means that the model's forward pass is mathematically identical before and after the adapter is merged. This is in direct contrast to the "lossy" merge of a standard LoRA adapter into quantized weights.
> > 1) **Lossy Merge**: As shown in Figure 2, merging a 16-bit LoRA adapter into quantized weights forces a difficult trade-off: either you sacrifice the adapter's precision to maintain low-bit inference (an accuracy loss), or you maintain the adapter's precision by de-quantizing the weights, which sacrifices inference efficiency (an efficiency loss).
> > 2) **Lossless Merge**: LoTA-QAF directly adjusts weights on the quantization grid and influences quantization parameters via its offset factor. Regarding the offset factor μ, its calculation is performed identically during both the training phase and the inference phase (after the merge). Since the model's computational logic remains consistent, the merge operation introduces no loss to the adapter's learned information, ensuring the adapter's effect is preserved from training to inference.
> > 3) **Cost**: We also wish to clarify that our method is not without its own trade-offs. The primary cost of LoTA-QAF is the limited representational capacity of its ternary adapters. While our method sacrifices the high precision of a 16-bit LoRA, the ternary adapters can be losslessly merged while still directly modifying the integer weights.
>
> ---
>
> > **Q2**: The proposed t-signSGD does not appear to be significantly different from the previously proposed SignSGD.
>
> > **A2**: We must emphasize that our t-SignSGD employs a fundamentally different learning process compared to classical SignSGD for the following reasons:
> > 1) **The update mechanism is different.** We do not use the standard learning_rate * sign(gradient) formula. Instead, t-SignSGD updates are determined by a dynamic percentile-based threshold $\sigma_t$ in Eq. (6), which focuses updates on the most prominent gradients. If a ternary adapter weight is selected for an update because its gradient magnitude exceeds the threshold $\sigma_t$, its value is adjusted directly by adding or subtracting 1 within the {-1, 0, 1} set. This threshold-based selection, rather than a learning rate, is the crucial factor determining the update's selectivity.
> > 2) **t-SignSGD is tightly integrated with the ternary adapter framework.** Such a large-step adjustment (+1 or -1) would be infeasible if applied directly to neural network parameters. Our approach is effective because it is specifically designed for the discrete and bounded nature of the ternary adapters. The updates from t-SignSGD do not directly modify the quantized weights. Instead, the updated ternary adapters are first multiplied to form an auxiliary matrix, which is then processed through Equations (3) and (4) to indirectly influence the quantized weights.
>
> ---
>
> > **Q3**: Limited evaluation: Evaluations on recent LLMs, such as Llama3, or on commonsense QA tasks were not conducted.
>
> > **A3**: The models we evaluated—Llama 3.1 8B (July 2024), Qwen 2.5 14B (Jan 2025), Qwen 2.5 32B (Jan 2025), and Llama 3.1 70B (Dec 2024)—were selected because they were benchmark models when we began our experimental phase. These details are available in Section 4.1 ("Models and Quantization") of our paper.
> >
> > Furthermore, we have supplemented our work with new experiments on the CommonsenseQA. We evaluated the performance of Llama 3.1 8B and Qwen 2.5 14B on this task.
> >
> > | Model | Mothed | 4-bit | 3-bit | 2-bit |
> >|:---|:---|:---:|:---:|:---:|
> >| **8B** | lora (x-bit +16-bit)  | 77.55 | 64.52 | 19.24 |
> >|  | lota (x-bit)  | 78.62 | 65.76 | 21.04 |
> >| **14B** | lora (x-bit +16-bit)  |83.06 | 81.71 | 32.19 |
> >|  | lota (x-bit)  | 83.70 | 82.96 | 35.95 |
> >
> > These results on CommonsenseQA are consistent with our findings on MMLU (performance-recovery fine-tuning task). This is because the primary goal is to restore the model's general knowledge degraded during quantization. Our method, LoTA-QAF, excels here as it is specifically designed to directly adjust weights on the quantization grid, which is effective for recovering these broad, pre-existing capabilities.

---

> > ### Comment · Reviewer_abLM · 2025-08-04
> > **Thanks for the rebuttal**
> >
> > Thank you for the authors' rebuttal. I especially appreciate their clarification of the trade-offs of their method and the key differences between t-sign SGD and sign SGD. I hope these clarifications will be appended to the final version. I keep my score.

---

> > > ### Author Response · Authors · 2025-08-05
> > >
> > > We sincerely thank you for your thoughtful review and are delighted that you appreciated our clarifications regarding the method's trade-offs and the distinctions between t-SignSGD and SignSGD. We confirm that these important clarifications will be integrated into the final manuscript to benefit all readers. Thank you once again for your constructive engagement.
> > >
> > > Best regards.

---

### Official Review · Reviewer_N8rD · 2025-07-02

**Clarity:** 2
**Significance:** 4
**Originality:** 3
**Rating:** 4
**Confidence:** 4

**Summary:**

The authors introduce LoTA-QAF, a PEFT method that directly tackles the challenge of merging adapters into low-bit base weights. They initialize the ternary adapter AB, and apply a threshold to their multiplication to make sure that it is also ternary. The ternary nature of the adapter allows for a lossless merging with the base INT weights through modifying both the quantized values and the zero point. They further introduce t-SignSGD, an optimizer tailored for ternary parameters. Extensive experiments show the effectiveness of the method.

**Questions:**

1. It’s not clear to me what the forward pass of the unmerged module actually is. Is it $(W_q + \frac{\alpha}{r} A_T B_T)^T x$, similar to LoRA? Or are you directly defining it as $(s \cdot W’_{int} + z’)^T x$ with no $\alpha$? I think it helps the reader to define it explicitly, and discuss the differences from LoRA forward pass.
2. Can you please elaborate on the definition of the residual matrix? From the term "residual", my intuition says it should be $\tilde{W}=\Delta W - \hat{W}$ as opposed to $\tilde{W}=\Delta W - \omega \hat{W}$. Can you please elaborate on your design and the intuition/reasoning behind it?
3. In Figure 3, why are the values that are 3 and -3 not included in $\hat{W}$? Is that because the corresponding values in $W_{int}$ are on the boundary? So the indices that are on the boundary are always zero in $\hat{W}$? I think this can confuse the reader if not explained explicitly.
4. LoTA-QAF is tailored specifically for INT base weights. Given the recent shift of the literature toward non-uniform quantized formats such as MXFP4, do you think your method could be extended to support those? A discussion on this would strengthen the paper.
5. In my opinion t-SignSGD is an interesting new optimizer. However, I don’t think it is studied extensively in the paper. For example, how do the parameters change during training, e.g., what percentage of them change in both directions during training? Can we say anything theoretical about its convergence (probably the rebuttal period is too short for that though)?
6. In the context of task-specific experiments, while LoTA-QAF shows strong results, it is still consistently and significantly outperformed by LoRA. I think that is fine, but it raises a question that is not answered in the paper: is there “any” 2/3/4 bit solution that recovers LoRA? In other words, can LoTA-QAF be improved, or is there a fundamental constraint that a standalone 2/3/4-bit weight simply cannot recover LoRA? I can think of two ways to approach this: (1) less interesting: merge the LoRA adapter into the weights in high precision, and then apply 4-bit SpinQuant/FlatQuant/Quarot [1,2,3]  and see if it improves upon LoTA-QAF, and (2) more interesting: run a weight-only version of HALO [4] to directly fine-tune a low-precision model (the activations should be kept in high precision). I think having these results, especially the second one, could significantly strengthen your paper.

In general, I think LoTA-QAF is a novel and strong method that has potential impact on the community. However, I think the writing on parts of the paper could be improved, and some more ablations/experiments (especially on t-SignSGD) are missing. Depending on how the rebuttal goes, I am open to increasing my score.

[1] https://arxiv.org/pdf/2405.16406
[2] https://arxiv.org/pdf/2410.09426
[3] https://arxiv.org/pdf/2404.00456
[4] https://arxiv.org/pdf/2501.02625

**Ethical Concerns:**

["NO or VERY MINOR ethics concerns only"]

**Final Justification:**

The writing on some parts could be improved and some experiments are not fully convincing. Generally though, I think this is a solid paper with potential impact on the community. Hence I raise my score to "borderline accept."

**Limitations:**

Yes

**Quality:**

3

**Strengths And Weaknesses:**

Strengths:
1. The idea is interesting and it is likely that the community will use this method.
2. The experiments are extensive and include speedup results.

Weaknesses (see Questions for more details):
1. To me, the method section was unclear, specifically Section 3.2 and Figure 3.
2. More discussion/ablations on t-SignSGD would strengthen the paper.
3. No discussion on non-uniform quantization formats.

---

> ### Author Rebuttal · Authors · 2025-07-29
>
> We thank the reviewer for recognizing the novelty of our idea and its potential value to the community. We will address your insightful questions to further clarify the technical details.
>
> > **Q1**: It’s not clear to me what the forward pass of the unmerged module actually is. Is it $(W_q + \frac{\alpha}{r} A_T B_T)^T x$, similar to LoRA? Or are you directly defining it as $(s \cdot W’_{int} + z’)^T x$ with no $\alpha$? ...
>
> > **A1**: Our method follows the same logic as lossless merge, $(s \cdot W’_{int} + z’)$. In contrast to LoRA, we do not use an $\alpha$ to scale the adaptation. Instead, the impact of the adaptation is controlled by a threshold $\omega$. As we briefly mentioned in Section 4.3, this threshold $\omega$ functions similarly to $\alpha$. It determines how the auxiliary matrix $\Delta \mathbf{W} = \mathbf{A}_T \mathbf{B}_T$ contributes to the ternary matrix $\mathbf{\hat{W}}$. A smaller $\omega$ allows the adapter to have a greater impact on the quantized weights, while a larger $\omega$ has a more conservative effect.
>
> ---
>
> > **Q2**: Can you please elaborate on the definition of the residual matrix? ...
>
> > **A2**: The auxiliary matrix $\Delta W$ is utilized to generate a ternary matrix $\mathbf{\hat{W}} \in \\{-1, 0, 1\\}$ and a offset factor. This is controlled by a threshold $\omega$. The rule is as follows: if the signal's magnitude $|\Delta W_{ij}|$ is greater than $\omega$, we set $\mathbf{\hat{W}}_{ij} = sign(\Delta \mathbf{W}\_{ij})$. The residual, which represents the signal that remains, is then calculated as: $\mathbf{\tilde{W}}\_{ij} = \Delta \mathbf{W}\_{ij} - \omega \mathbf{\hat{W}}\_{ij}$.
>
> ---
>
> > **Q3**: In Figure 3, why are the values that are 3 and -3 not included in $\hat{W}$? ...
>
> > **A3**: We perform a boundary check to prevent overflow or underflow of the quantized weights, but this does not mean that updates in $\hat{W}$ for weights on the boundary are always zero. For example, a weight of `15` can still be updated by `-1`, just not by `+1`. In Figure 3, the potential updates from `3` and `-3` in $\Delta W$ are blocked because their corresponding weights in $W_{int}$ are at the boundaries (`15` and `0`), and the updates would have caused an overflow.
>
> ---
>
> > **Q4**: Given the recent shift of the literature toward non-uniform quantized formats such as MXFP4, ...
>
> > **A4**: LoTA-QAF can be extended to support non-uniform quantization formats such as MXFP4 or NF4, though it would require a more sophisticated design for the threshold $\omega$.
> >
> > The core idea is that both uniform and non-uniform formats map high-precision values to a discrete set of representation points (e.g., 16 unique values for a 4-bit format). For non-uniform formats, instead of an arithmetic `+/- 1` adjustment, we can define the adjustment as moving to an adjacent representation level (e.g., `0010` -> `0011`).
> >
> > The main challenge is that the step sizes between adjacent levels are not uniform. The value change from `0010` -> `0011` differs from `0011` -> `0100`. Consequently, a fixed adjustment threshold $\omega$ is no longer optimal. The threshold would need to be adaptive, depending on the specific step size of a potential change. This is an interesting direction, and we thank you for raising it.
>
>
> ---
>
> > **Q5**: In my opinion t-SignSGD is an interesting new optimizer. However, I don’t think it is studied extensively in the paper. ...
>
> > **A5**: We sincerely thanks for your positive feedback on t-SignSGD and your insightful questions regarding its convergence properties. Our proposed t-SignSGD can be understood as a gradient-guided discrete search algorithm whose design includes several mechanisms to ensure convergence.
> >
> > We will elaborate on this from three perspectives: 1) The Challenge of Formal Convergence Analysis, 2) Theoretical Grounding in SignSGD, and 3) Analysis of the t-SignSGD Update Rule for Convergence.
> >
> > *Please refer to Qp8Z-A2 for the full details on these three perspectives, given the space limitations here.*
>
>
> > **Experiment on Ternary Adapter Weight Changes under t-SignSGD Training**
> >
> > To better understand the dynamics of our optimizer, we analyzed the parameter changes of the ternary adapters for Llama 3.1 8B after fine-tuning on the Alpaca dataset. The results are as follows:
> >
> > | Metric | Value | Percentage | Description |
> > |---|---|---|---|
> > | Total Parameters | 167,772,160 | 100% | Total adapter parameters across 32 layers. Adapters are applied to 7 key projections within the attention and MLP blocks of each layer. |
> > | Increased Parameters | 28,952,566 | 17.26% | Count of weights that changed in the positive direction. |
> > | Decreased Parameters | 28,960,070 | 17.26% | Count of weights that changed in the negative direction . |
> > | Unchanged Parameters | 109,859,524 | 65.48% | Count of weights that are unchanged. |
> >
> > From this experiment, we can learn that the rate of change in the adapters is nearly identical in both positive and negative directions (17.26% vs. 17.27%). This suggests that the adaptation is not a simple unidirectional shift but a balanced, bidirectional fine-tuning process to fit the target task.
> >
> > One-third of the parameters changed during adaptation, while two-thirds remained unchanged. This phenomenon is a result of the t-SignSGD design, specifically the adaptive threshold $\mathbb{I}_{|g_t| > \max(\tau, \sigma_t)}$. This mechanism ensures our updates are based on a large gradient magnitude, which implies a high probability that the sign is correct. Conversely, it filters out updates associated with small gradient signals, which can be noisy or ill-scaled.
> ---
>
> > **Q6**: In the context of task-specific experiments, while LoTA-QAF shows strong results, it is still consistently and significantly outperformed by LoRA. I think that is fine, but it raises a question that is not answered in the paper: ...
>
> > **A6**: To build intuition, let's first consider the process of model quantization, separate from fine-tuning. One can visualize quantization as squeezing the air out of a plastic bottle to fill it completely with water. A more thoroughly trained model contains more "water" (knowledge). When squeezing out the "air" (parameter redundancy), it becomes easier to accidentally squeeze out some "water" as well, causing performance degradation.
> >
> > If we only consider adjusting the quantized weights (and not the quantization parameters), it is possible to match or closely approximate LoRA's performance. This is observable on tasks where the model is already proficient or on simpler tasks (e.g., Performance-Recovery Fine-Tuning or even GSM8K). Furthermore, if the model size is large enough, like 70B, the task-specific performance will also be closer to LoRA's, because the model has a sufficient number of parameters to adjust. To extend the analogy, if fine-tuning is like reshaping the plastic bottle into a specific form, it can be done perfectly if the bottle is very soft (a large model) or the target shape is simple (an easy task).
> >
> > When considering methods like LoTA-QAF, which adjust both quantized weights and quantization parameters, the question becomes even more complex, as there are more ways to "reshape the bottle." Therefore, it is difficult to definitively state whether "any 2/3/4-bit solution can recover LoRA" without first specifying the model, task, and method.
> However, we agree this is an important and interesting question. We conducted an experiment by applying Quarot to re-quantize the result of the GPTQ+LoRA method on the Llama 3.1 8B model.
> >
> > | Bit | Method | GSM8k | SQL | ViGGO |
> > | :--- | :--- | :---: | :---: | :---: |
> > | **4-bit** | GPTQ+LoRA | 70.20 | 76.70 | 88.64 |
> > | | Quarot (GPTQ+LoRA) | 67.53 | 74.12 | 48.54 |
> > | | QA-LoRA | 68.69 | 74.10 | 51.15 |
> > | | LoTA | 70.05 | 74.60 | 60.30 |
> > | **3-bit** | GPTQ+LoRA | 65.66 | 75.80 | 82.20 |
> > | | Quarot (GPTQ+LoRA) | 60.52 | 71.64 | 41.89 |
> > | | QA-LoRA | 62.17 | 72.20 | 43.86 |
> > | | LoTA | 63.53 | 73.10 | 55.96 |
> > | **2-bit** | GPTQ+LoRA | 34.12 | 72.50 | 80.70 |
> > | | Quarot (GPTQ+LoRA) | 15.92 | 52.17 | 34.07 |
> > | | QA-LoRA | 17.36 | 56.20 | 37.18 |
> > | | LoTA | 19.48 | 67.00 | 41.98 |
> >
> > The results show that applying post-training quantization (PTQ) after fine-tuning is less effective than fine-tuning directly on quantized weights. This observation persists even when using an advanced PTQ method like Quarot, which is specifically designed to handle outliers. The performance degradation from re-quantization is particularly severe in the 2-bit setting. This highlights the necessity of developing QAF-type methods like LoTA-QAF.
> >
> > As for using HALO to test the performance ceiling in low-bit settings, this is an avenue worth exploring. However, given that HALO is not directly adapted for 2/3/4-bit training, and achieving optimal performance would require extensive experimentation, we did not conduct this experiment.
> ---
> Thank you again for your constructive review. We hope our responses have clarified your concerns and further demonstrated the potential of LoTA-QAF.

---

> > ### Comment · Reviewer_N8rD · 2025-08-04
> >
> > I would like to thank the authors for the additional discussions and new experiments. Regarding t-SignSGD and the thresholding mechanism: while the analysis is not theoretically rigorous, I find the approach interesting, reasonable, and well-justified for the setting considered. More broadly, I believe LoTA-QAF has the potential to be a valuable contribution to the community.
> >
> > That said, I believe the writing of the paper requires improvement. For example, as noted by other reviewers, the claim of the method being “lossless” is not sufficiently elaborated upon, and the decision to skip updates when the value is on the boundary is not clearly stated, aside from being vaguely implied in a figure, which is potentially confusing.
> >
> > Additionally, as I previously noted, the performance drops compared to LoRA on fine-tuning tasks are substantial in some cases. While I appreciate the authors’ discussion ("plastic bag" metaphor) to acknowledge inevitable information loss, I still believe LoTA-QAF may be underperforming relative to what is possible in the 4-bit regime, and the current results are not sufficient to suggest otherwise.
> >
> > All in all, based on the authors’ rebuttal, I am increasing my score.

---

> > > ### Author Response · Authors · 2025-08-05
> > >
> > > We sincerely thank you for your thorough and insightful review of LoTA-QAF. We are greatly encouraged by your constructive feedback and positive assessment. To enhance the clarity and precision of our paper, we will revise the manuscript by providing a more detailed elaboration on the “lossless” claim, explaining the mechanism for skipping updates when a value is on the boundary, and addressing the other issues you previously noted.
> > >
> > > Regarding the performance of LoTA-QAF in the 4-bit regime, we plan to explore this further in our future work to better determine the optimal performance achievable by QAF methods in low-bit settings.
> > >
> > > Finally, we appreciate the opportunity to discuss our work with you. Your concerns have been invaluable, helping us to not only better present our findings but also to gain new perspectives from which to analyze our work.
> > >
> > > Best regards.

---

### Official Review · Reviewer_HKvF · 2025-07-03

**Clarity:** 3
**Significance:** 3
**Originality:** 3
**Rating:** 4
**Confidence:** 4

**Summary:**

To address (1) the mismatch between low-precision quantized weights and high-precision adaptation weights that constrains the inference efficiency of quantized weights, and (2) potential performance degradation from approximating high-precision adaptation weights during merging, the paper presents lossless ternary adaptation for quantization-aware fine-tuning (LoTA-QAF), which utilizes ternary adaptation weights and ternary signed gradient descent (t-SignSGD) for updating them. The paper shows the effectiveness of LoTA-QAF for various models (Llama, Qwen) and benchmarks (MMLU, GSM8K).

**Questions:**

Please see Strengths and Weaknesses

**Ethical Concerns:**

["NO or VERY MINOR ethics concerns only"]

**Final Justification:**

I maintain my positive score, 4.

**Limitations:**

Please see Strengths and Weaknesses

**Paper Formatting Concerns:**

N.A.

**Quality:**

3

**Strengths And Weaknesses:**

Strengths

It seems that the authors effectively address the issue mentioned in Summary, which is novel.

(1) They confirm that  LoTA-QAF can be about twice faster than LoRA after adapters are merged into quantized weights.

(2) LoTA-QAT consistently outperforms GPTQ+LoRA on MMLU in Table 1.

(3) The paper is well written.

Weaknesses

(1) One thing I am most concerned about is that in Table 1, LoTA-QAT performs better than GPTQ+LoRA on MMLU but worse than GPTQ+LoRA on GSM8K, which is likely to imply that LoTA-QAT is prone to overfitting or beneficial for some limited tasks only.

(2) Given that the authors use Alpaca, which is instruction-following data, it would be more appropriate to evaluate on GSM8K with few-shot, not zero-shot.

---

> ### Author Rebuttal · Authors · 2025-07-29
>
> We thank the reviewer for recognizing the novelty and effectiveness of our method. We will address your insightful questions to further clarify the technical details.
>
> > **Q1**: One thing I am most concerned about is that in Table 1, LoTA-QAT performs better than GPTQ+LoRA on MMLU but worse than GPTQ+LoRA on GSM8K, which is likely to imply that LoTA-QAT is prone to overfitting or beneficial for some limited tasks only.
>
> > **A1**: This performance difference is not an indication of overfitting or limited task applicability, but rather a direct reflection of the distinct objectives of the two fine-tuning paradigms.
> > 1) **Performance-recovery fine-tuning (MMLU):** On broad benchmarks like MMLU, the primary goal is to restore the model's general, pre-existing knowledge degraded during quantization. LoTA-QAF excels in this area as it is specifically designed for quantized models, directly adjusting weights on the quantization grid and influencing quantization parameters. This performance-recovery scenario requires broad-based adjustments to the model's parameters, rather than the highly specialized updates needed for learning new patterns. Consequently,  LoTA-QAF excels in performance-recovery fine-tuning.
> 2) **Task-specific fine-tuning (GSM8K):** Conversely, a task like GSM8K requires the model to specialize in a new, complex, and fine-grained domain (i.e., mathematical reasoning, we use 7.47k GSM8K training data). In this task-specific scenario, the higher precision and representational capacity of a 16-bit LoRA adapter provide a natural advantage in capturing the intricate patterns of this specialized domain.
> However, the key advantage of LoTA-QAF is that its adapters can be losslessly merged, preserving both the final accuracy and the full computational efficiency of the low-bit model for deployment. The superior performance of GPTQ+LoRA on GSM8K comes at the cost of inference inefficiency, as the 16-bit adapter cannot be losslessly merged.
>
> ---
>
> > **Q2**: Given that the authors use Alpaca, which is instruction-following data, it would be more appropriate to evaluate on GSM8K with few-shot, not zero-shot.
>
> > **A2**: We'd like to clarify that the Alpaca dataset is used for performance-recovery fine-tuning (on MMLU), not for the GSM8K task.
> For task-specific fine-tuning on GSM8K, we use its training set of 7.47k samples and test with 1.32k samples, as detailed in Section 4.1 ("Tasks"). Therefore, we chose a zero-shot setting to purely evaluate the reasoning capabilities acquired during this task-specific fine-tuning phase.
> Additionally, we provide the 5-shot results on GSM8K for the Llama 3.1 8B and Qwen 2.5 14B:
> >
> > | Model | Mothed | 4-bit | 3-bit | 2-bit |
> >|:---|:---|:---:|:---:|:---:|
> >| **8B (shot-5)** | lora (x-bit +16-bit)  | 74.53 | 68.55 | 37.72 |
> >|  | lota (x-bit)  | 73.61 | 66.94 | 23.63 |
> >| **14B (shot-5)** | lora (x-bit +16-bit)  | 85.21 | 82.25 | 57.85 |
> >|  | lota (x-bit)  | 84.03 | 80.38 | 55.82 |
> >| **8B (shot-0)** | lora (x-bit +16-bit)  | 70.20 | 65.66 | 34.12 |
> >|  | lota (x-bit)  | 70.05 | 63.53 | 19.48 |
> >| **14B (shot-0)** | lora (x-bit +16-bit)  | 80.06 | 72.18 | 37.23 |
> >|  | lota (x-bit)  | 78.37 | 70.36 | 36.25 |
> On GSM8K, using a 5-shot setting significantly improves performance, particularly for the 14B 2-bit model. This demonstrates that through LoRA/LoTA fine-tuning, the quantized models have effectively learned task-specific patterns via the adapters. When augmented with 5-shot examples, their performance is further boosted.

---

### Note · Authors · 2025-08-12

Dear Area Chair and Reviewers,

We sincerely thank you for your time and constructive evaluation. We are greatly encouraged that the reviewers acknowledged our paper's core contributions. They highlighted our approach as **novel and interesting** [HKvF, N8rD, Qp8Z], and recognized its **effectiveness**, supported by **extensive experiments** on large-scale models [HKvF, N8rD, abLM]. Furthermore, we are pleased the reviewers found the paper **well-written** [HKvF, abLM] and appreciated its potential as a **valuable tool for the community**, adaptable to diverse hardware and deployment constraints [N8rD, Qp8Z].

In our rebuttal, we provided detailed clarifications to address the reviewers' valuable questions. We confirm that these points will be integrated into the final manuscript to improve its clarity and completeness:

- **Clarification on "Lossless Merging" and Method Details**: We offered a more in-depth explanation of "lossless merging" and clarified mechanism details, such as the handling of boundary values during updates, to resolve questions noted by Reviewer N8rD and abLM.

- **Elaboration on t-SignSGD**: We detailed the principles of t-SignSGD and clarified its distinctions from the standard SignSGD, addressing the queries from Reviewer N8rD, abLM, and Qp8Z.

- **Deeper Performance Analysis**: We provided a richer discussion on the performance across different bit-widths, model sizes, and tasks, particularly addressing the trade-offs between our approach and full-precision adapters like LoRA, a point raised by Reviewer HKvF, N8rD, and Qp8Z.

- **Other Clarifications**: We will also ensure that all other questions and suggestions raised during the discussion are addressed throughout the final manuscript to enhance its overall quality.

Thank you again for your constructive engagement that has helped strengthen our work.

Best regards.

---

### Decision · Program_Chairs · 2025-09-17

**Decision:**

Accept (poster)

**Comment:**

This paper proposes LoTA-QAF. The approach is novel, with strong results across multiple models and bit-widths, often outperforming QA-LoRA and GPTQ+LoRA. The introduction of t-SignSGD is also interesting and empirically supported. Some remaining concerns include the notion of “lossless” merging not being fully clarified in the original paper, task-specific results still lag behind full-precision LoRA, and parts of the method section (e.g., Figure 3) could be clearer. Overall, despite these issues, the work made sufficient contributions to the community, and I recommend acceptance.